EMBO
Molecular Medicine

# Niacin-mediated Tace activation ameliorates CMT neuropathies with focal hypermyelination

Alessandra Bolino[1,2,*], Françoise Piguet[1,2,†], Valeria Alberizzi[1,2], Marta Pellegatta[1,2], Cristina Rivellini[1,2], Marta Guerrero-Valero[1,2], Roberta Noseda[1,2], Chiara Brombin[3], Alessandro Nonis[3], Patrizia D'Adamo[2], Carla Taveggia[1,2] & Stefano Carlo Previtali[1,2,4]

## Abstract

Charcot–Marie–Tooth (CMT) neuropathies are highly heterogeneous disorders caused by mutations in more than 70 genes, with no available treatment. Thus, it is difficult to envisage a single suitable treatment for all pathogenetic mechanisms. Axonal Neuregulin 1 (Nrg1) type III drives Schwann cell myelination and determines myelin thickness by ErbB2/B3-PI3K–Akt signaling pathway activation. Nrg1 type III is inhibited by the α-secretase Tace, which negatively regulates PNS myelination. We hypothesized that modulation of Nrg1 levels and/or secretase activity may constitute a unifying treatment strategy for CMT neuropathies with focal hypermyelination as it could restore normal levels of myelination. Here we show that *in vivo* delivery of Niaspan, a FDA-approved drug known to enhance TACE activity, efficiently rescues myelination in the *Mtmr2*[−/−] mouse, a model of CMT4B1 with myelin outfoldings, and in the *Pmp22*[+/−] mouse, which reproduces HNPP (hereditary neuropathy with liability to pressure palsies) with tomacula. Importantly, we also found that Niaspan reduces hypermyelination of *Vim* (vimentin)[−/−] mice, characterized by increased Nrg1 type III and Akt activation, thus corroborating the hypothesis that Niaspan treatment downregulates Nrg1 type III signaling.

**Keywords** animal models; Charcot–Marie–Tooth neuropathies; myelin; Neuregulin 1; nicotinic acid

**Subject Categories** Genetics, Gene Therapy & Genetic Disease; Neuroscience

## Introduction

Charcot–Marie–Tooth (CMT) neuropathies have a collective prevalence of 1:2,500, and as a whole represent the most common form of human hereditary neuromuscular disease. CMTs are highly heterogeneous disorders commonly characterized by progressive muscular weakness, atrophy, and sensory loss (Pareyson & Marchesi, 2009; Rossor *et al*, 2013; Saporta & Shy, 2013). Symptoms progress in a length-dependent manner and constitute an important cause of disability with high social costs. Although CMTs can be primarily demyelinating or axonal, at later stages both components are affected and disability parallels axonal loss. Notably, CMTs with primary myelin involvement comprise forms with reduced myelin thickness at birth, such as congenital hypomyelination, or demyelinating CMT, thus following normal myelin formation, and those with excessive-redundant myelin thickness, such as myelin outfoldings, hypermyelination, and/or tomacula.

CMTs are due to mutations in at least 70 different genes, and the analyses of the underlying molecular mechanisms have revealed their highly heterogeneous pathogenesis. Given the high and increasing number of distinct CMT subtypes, it is plausible and desirable to envisage unifying therapies to treat CMT neuropathies. However, at present, no known therapy is available for any CMT neuropathy.

Neuregulin 1 (Nrg1) type III, a member of the Nrg1 family of proteins, is an essential instructive signal for peripheral myelination, which drives Schwann cell binary choice between myelination and non-myelination (Taveggia *et al*, 2005). Importantly, the amount of axonal Nrg1 type III determines the thickness of the myelin sheath (Michailov *et al*, 2004; Taveggia *et al*, 2005). Binding of Nrg1 to their cognate receptors ErbB2/B3 on Schwann cell plasma membrane activates the PI3K–Akt signaling pathway (Taveggia *et al*, 2005). Nrg1 type III activity is regulated by the extracellular cleavage of secretases. While the β-secretase Bace1 activates Nrg1 type III, enhancing myelination (Hu *et al*, 2006; Willem *et al*, 2006), the α-secretase Tace cleavage of Nrg1 type III inhibits myelination (La Marca *et al*, 2011). Accordingly, mutant mice lacking neuronal Tace are hypermyelinated and their phenotype remarkably resembles Nrg1 type III overexpressing mice (Michailov *et al*, 2004; La Marca *et al*, 2011).

We postulated that modulation of Tace activity may constitute a unifying treatment strategy for hypermyelinating CMTs as it could

1   INSPE-Institute of Experimental Neurology, San Raffaele Scientific Institute, Milan, Italy
2   Division of Neuroscience, San Raffaele Scientific Institute, Milan, Italy
3   University Centre of Statistics in the Biomedical Sciences (CUSSB), Vita-Salute San Raffaele University, Milan, Italy
4   Department of Neurology, San Raffaele Scientific Institute, Milan, Italy
    *Corresponding author. Tel: +39 02 26364743; Fax: +39 02 26435093; E-mail: bolino.alessandra@hsr.it
    †Present address: Institut de Génétique et de Biologie Moléculaire et Cellulaire (IGBMC), Strasbourg, France

restore myelination and likely preserve nerve physiology. Here we report that Niaspan, an extended release formulation of nicotinic acid/niacin, which is thought to enhance Tace activity (Chen *et al*, 2007, 2009), ameliorates the neuropathy in the *Mtmr2* (myotubularin-related protein 2)$^{-/-}$ mouse, a model of CMT4B1 with myelin outfoldings (Bolino *et al*, 2000, 2004; Bolis *et al*, 2005), and in the *Pmp22* (peripheral myelin protein 22)$^{+/-}$ mouse, which reproduces HNPP (hereditary neuropathy with liability to pressure palsies) with tomacula (Adlkofer *et al*, 1995). Importantly, we also found that niacin does not rescue hypermyelination in *Tace*$^{-/-}$ co-culture models, whereas Niaspan treatment reduces hypermyelination of *Vim* (vimentin)$^{-/-}$ mice, characterized by increased Nrg1 type III and Akt activation (Triolo *et al*, 2012). Altogether, these data corroborate the hypothesis that Niaspan treatment ameliorates myelination in neuropathic models by enhancing Tace activity and downregulating Nrg1 type III signaling.

# Results

### Nrg1 type III signaling pathway in the *Mtmr2*$^{-/-}$ mouse, a model of CMT4B1 neuropathy with myelin outfoldings

We postulated that niacin/Niaspan (nicotinic acid), by increasing Tace (Tumor necrosis factor-α converting enzyme) activity and downregulating Nrg1 type III, could ameliorate focal hypermyelination, prevent myelin degeneration, and preserve axonal physiology. In the perspective of a preclinical application of this strategy, we first characterized the expression profile of Tace during nerve development. Tace is expressed in both Schwann cells and axons, but myelination is regulated by axonal Tace (La Marca *et al*, 2011). To determine Tace expression profile in axons, we generated *Tace*$^{Fl/Fl}$; *P0-Cre* mice, in which the *MPZ* (myelin protein zero) promoter drives Cre recombinase expression specifically in Schwann cells, starting from E13.5 (La Marca *et al*, 2011). In this mutant, Tace expression is specifically downregulated in Schwann cells. By comparing *Tace*$^{Fl/Fl}$;*P0-Cre* and wild-type nerve lysates, we noted that Tace is also expressed in axons although at lower levels as compared to Schwann cells (Fig 1A). To note that in the nerve, Tace is detected as two main bands or isoforms, of which the higher of approximately 110 kDa is thought to be inactive as not yet processed by furin cleavage (Gooz, 2010), whereas the lower, of 80 kDa, should correspond to the fragment generated by furin cleavage (Fig 1B, wild-type rat nerve lysate). Axonal Tace expression is modulated in postnatal nerve development from P5 to P60, with a progressive decline around P20 (Fig 1B).

To assess the efficacy of our strategy, we first explored autosomal recessive CMT4B1 neuropathy, characterized by demyelination with childhood onset and myelin outfoldings (Previtali *et al*, 2007). We previously reported that loss of the MTMR2/Mtmr2 phospholipid phosphatase causes CMT4B1 in humans and mouse, and proposed myelin outfoldings as a model of altered membrane homeostasis in Schwann cells (Bolino *et al*, 2000, 2004; Bolis *et al*, 2005, 2009). Interestingly, recent studies have hypothesized that myelin outfoldings in the nerve might arise as a consequence of increased PIP$_3$ (phosphatidylinositol-3,4,5-triphosphate) levels and/or enhanced Akt/mTOR pathway activation (Goebbels *et al*, 2012; Domenech-Estevez *et al*, 2016). Thus, we first investigated Tace expression and

the Nrg1/ErbB2 pathway in sciatic nerves and Schwann cell/DRG neuron co-cultures from *Mtmr2*$^{-/-}$ mice, which reproduce myelin outfoldings (Bolis *et al*, 2009). We observed a modest increase in Tace expression levels in *Mtmr2*$^{-/-}$ adult nerves, suggesting that in this mutant there might be a physiological attempt to lower excessive myelination (Fig 1C and D). Western blot analysis did not show differences in Nrg1 type III expression levels and in Akt (murine thymoma viral oncogene homolog) and Erk (mitogen activated-protein kinase) phosphorylation in *Mtmr2*$^{-/-}$ sciatic nerves at P10, P20, and P60 (Fig EV1). Moreover, in *Mtmr2*$^{-/-}$ Schwann cell/DRG neuron co-cultures after 7 and 13 days of ascorbic acid treatment, Nrg1 type III expression levels and phosphorylation of Akt and Erk were also similar to controls (Figs EV2 and EV3). Finally, ErbB2/B3 receptor phosphorylation was significantly increased in *Mtmr2*$^{-/-}$ sciatic nerves at P2 but not at P10 and P15 (Fig 1E). Phosphorylation of ErbB2 receptors was also increased in *Mtmr2*$^{-/-}$ myelin-forming explants after 4 days of ascorbic acid treatment although the difference between mutant and control cultures was not statistically significant (Fig 1F). These data might suggest that the regulation of ErbB2 receptor trafficking is impaired in *Mtmr2*$^{-/-}$ Schwann cells, which in turn may result in a transient and local increase in signaling pathways relevant for PNS myelination.

### Downregulation of Nrg1 type III signaling reduces myelin outfoldings both *in vitro* and *in vivo*

To provide proof of principle of our strategy, we downregulated Nrg1 type III signaling in *Mtmr2*$^{-/-}$ co-culture explants using different strategies. First, we produced lentiviral vectors (LVs) expressing Nrg1 type III shRNA, which were validated in isolated rat neurons and in myelin-forming mouse explants (Fig 2A–D). We found that Nrg1 type III shRNA LVs downregulated the PI3K–Akt pathway and efficiently rescued myelin outfoldings (Fig 2E). Similarly, rhTACE (recombinant human TACE) treatment of *Mtmr2*$^{-/-}$ co-cultures decreased Nrg1 type III and Akt phosphorylation levels and rescued myelin outfoldings (Fig 3A and B). Next, we treated *Mtmr2*$^{-/-}$ co-cultures using niacin, nicotinic acid, which is known to enhance Tace activity. Consistent with our hypothesis, we observed that niacin treatment increased Tace activity and efficiently rescued myelin outfoldings in *Mtmr2*$^{-/-}$ cultures (Fig 3C and D). To confirm that Tace is the specific target of niacin, we performed two different experiments. First, we used *Tace*$^{-/-}$ explants, which produce more myelin segments than controls due to the loss of Tace-mediated inhibition on Nrg1 type III signaling and myelination (La Marca *et al*, 2011). We observed that niacin did not restore normal myelination levels in *Tace*$^{-/-}$ culture explants (Fig 3E). Second, we downregulated *Tace* expression in *Mtmr2*$^{-/-}$ co-cultures by means of shRNA LV transduction (La Marca *et al*, 2011) and we found that niacin did not rescue myelin outfoldings in *Mtmr2*$^{-/-}$ co-cultures with reduced Tace expression (Fig EV4). Altogether, these data confirm that the effect of niacin on myelination is mediated by the modulation of Tace activity.

Finally, to prove efficacy of our strategy *in vivo*, we genetically reduced Nrg1 type III levels by generating *Mtmr2*$^{-/-}$;*Nrg1*$^{+/-}$ mice. Morphological analyses at 6 months showed a significant reduction in myelin outfoldings in *Mtmr2*$^{-/-}$;*Nrg1*$^{+/-}$ as compared to *Mtmr2*$^{-/-}$ mice (Fig 4A). We also observed that *Mtmr2*$^{-/-}$;*Nrg1*$^{+/-}$, and *Nrg1*$^{+/-}$ sciatic nerves had similar g-ratio values and Akt

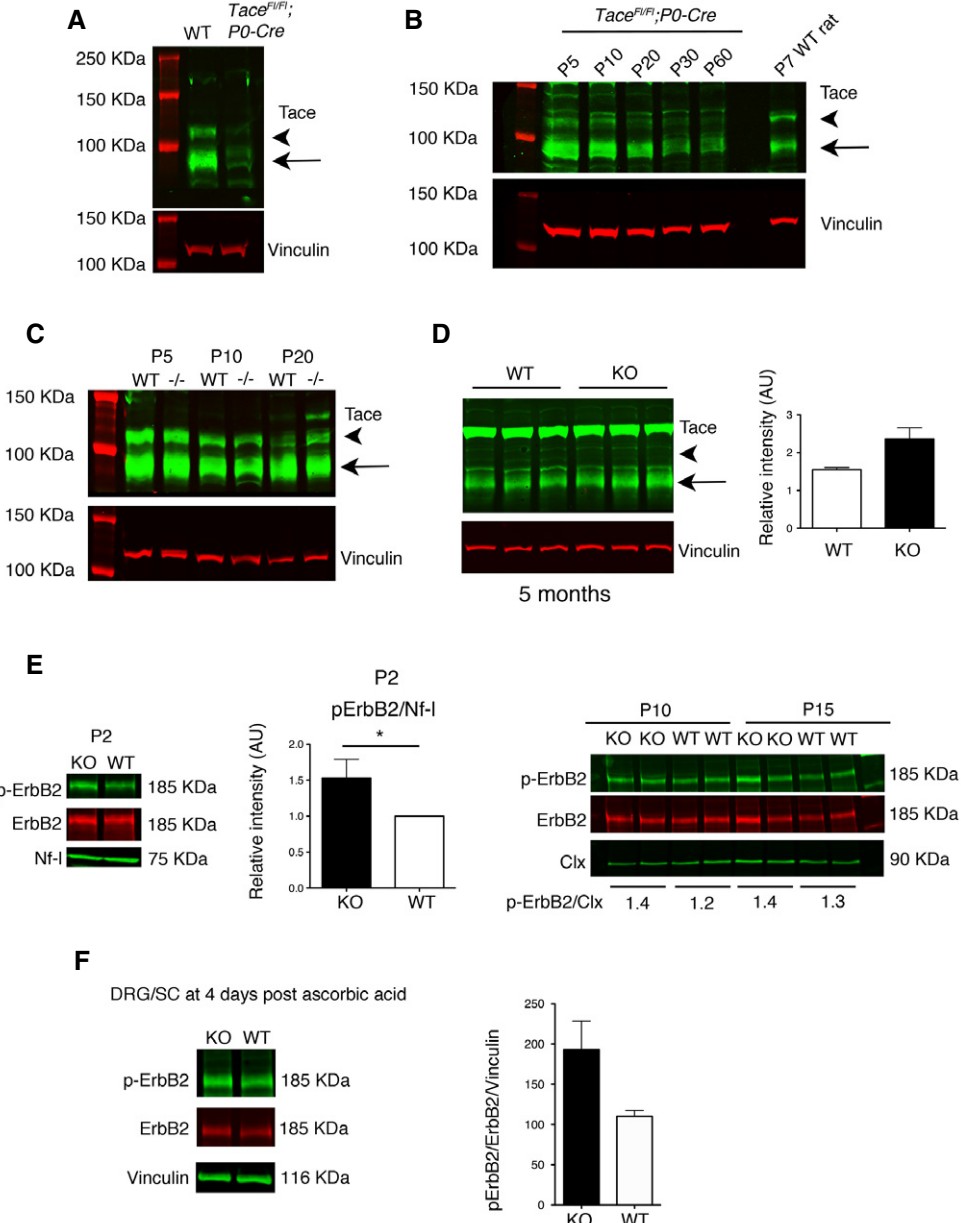

**Figure 1.  Expression levels of Tace and phosphorylation of ErbB2 in *Mtmr2*$^{-/-}$ sciatic nerves and Schwann cell/DRG neuron co-cultures.**

A   Western blot analysis of Tace using lysates from WT and *Tace*$^{Fl/Fl}$;*P0-Cre* mouse sciatic nerves at P10 (65 μg of total protein lysate loaded). Representative of two independent experiments.

B   Western blot analysis (50 μg of total protein lysate loaded) shows that Tace expression in axons is maximal at P5-P10 and then progressively declines in the adult. Representative of two independent experiments.

C   Western blot analysis of Tace in *Mtmr2*$^{-/-}$ sciatic nerves at P5, P10, and P20. Representative of two independent experiments.

D   Western blot analysis of Tace using lysates from *Mtmr2*$^{-/-}$ mouse sciatic nerves at 5 months, with quantification, *P* = 0.10, two-tailed Mann–Whitney *U*-test. Representative of two independent experiments.

E   Phosphorylation of ErbB2 in *Mtmr2*$^{-/-}$ sciatic nerves at P2, P10, and P15. At P2, each lane is a pool of *n* = 7 animals per genotype, representative of six independent experiments using *n* = 6 different nerve pools per genotype, and quantification using WT values arbitrary set to 1. *$P$ = 0.0313, Wilcoxon rank-sum test. At P10 and P15, mean values of two samples are shown. At P10, each lane is a pool of *n* = 3/4 animals per genotype, representative of three independent experiments using *n* = 6 different pools per genotype. Nf-l, neurofilament light chain; Clx, calnexin.

F   Phosphorylation of ErbB2 in *Mtmr2*$^{-/-}$ Schwann cell/DRG neuron co-cultures after 4 days of ascorbic acid treatment. Each lane is a pool from at least 10 coverslips/DRG per genotype. Representative of two independent experiments using *n* = 4 different pools of coverslips/DRG per genotype, *P* = 0.20, two-tailed Mann–Whitney *U*-test.

Data information: Results in (D–F) are expressed as mean ± SEM. In (A–D), arrowheads indicate the pro-protein and arrows show the cleaved active form of Tace. Source data are available online for this figure.

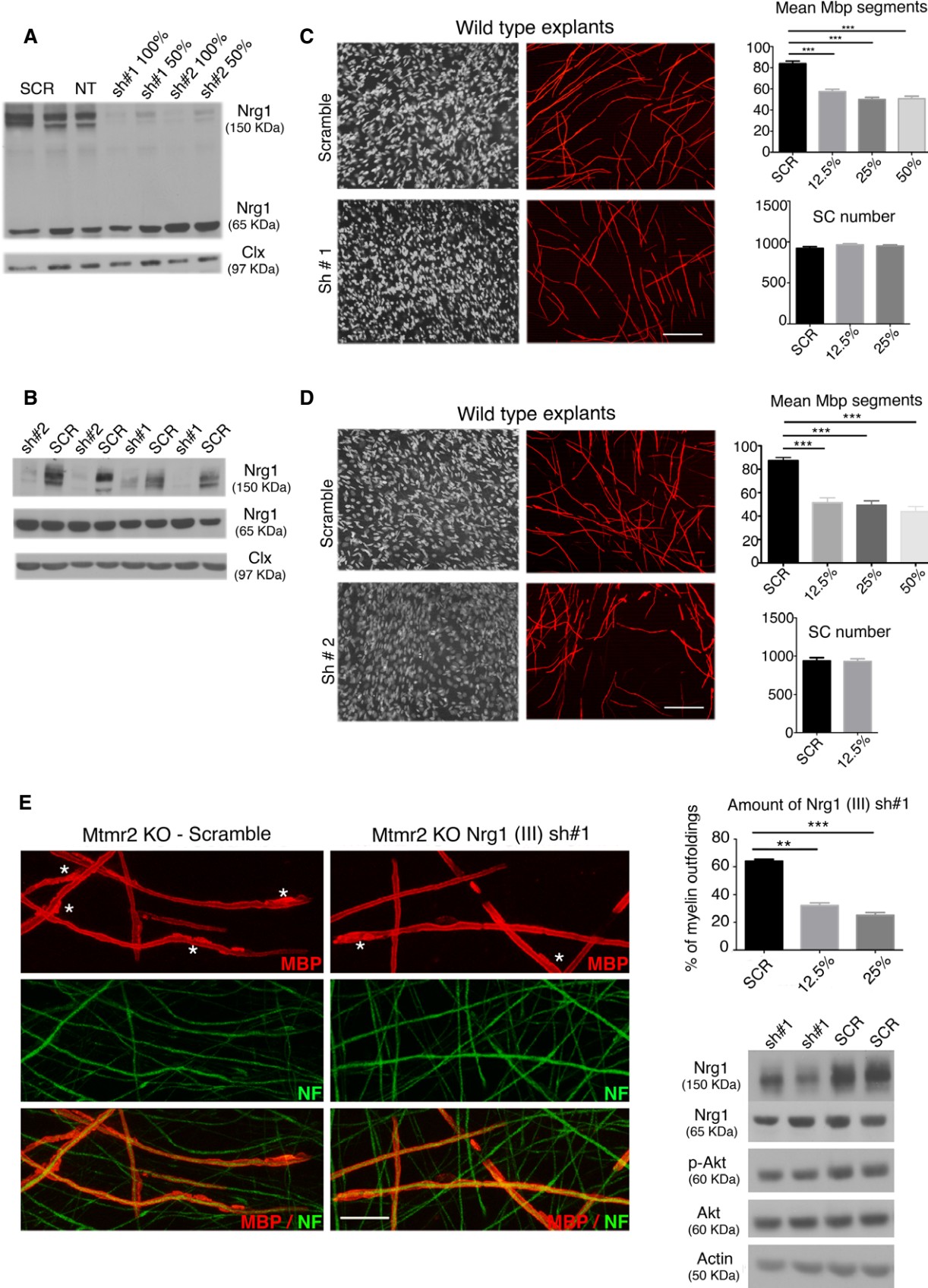

Figure 2.

**Figure 2.  Nrg1 type III shRNA lentiviral vector transduction reduces myelin outfoldings in *Mtmr2*$^{-/-}$ cultures.**

A    shRNA targeting Nrg1 type III were validated in rat purified neurons. Western blot analysis for Nrg1 shows reduction in its expression (pro-protein at 150 kDa) using two different hairpins, representative of two independent experiments. Clx, calnexin.

B    shRNA #1 and 2 downregulate Nrg1 type III expression also in myelin-forming Schwann cell/DRG neuron co-cultures analyzed after 7 days of ascorbic acid treatment, as shown by Western blot analysis, representative of two independent experiments. Clx, calnexin.

C, D  Titration of shRNA #1 (C) and of shRNA #2 (D) LVs (lentiviral vectors) on WT Schwann cell/DRG neuron co-cultures analyzed after 7 days of ascorbic acid treatment, with quantification of the mean number of Mbp-positive segments and Schwann cell number. In (C) $n = 20$ (SCR, scramble), $n = 11$ (LV sh#1, 12.5%), $n = 16$ (LV sh#1, 25%), $n = 9$ (LV sh#1, 50%) DRG/coverslips from two independent experiments. $P < 0.0001$, nonparametric one-way ANOVA, followed by Dunn's *post hoc* test. In (D) $n = 14$ (SCR), $n = 10$ (LV sh#2, 12.5%), $n = 12$ (LV sh#2, 25%), $n = 79$ (LV sh#2, 50%) DRG/coverslips, representative of two independent experiments. $P < 0.0001$, nonparametric one-way ANOVA, followed by Dunn's *post hoc* test. Scale bars, 100 μm.

E    Representative confocal images of *Mtmr2*$^{-/-}$ co-cultures transduced using LV expressing Nrg1 type III shRNA #1, with quantification of the percentage of myelin outfoldings, $n$ = number of DRG/coverslips, $n = 9,382$ fibers (SCR), $n = 9,368$ fibers (LV sh#1, 12.5%), $n = 8,327$ fibers (LV sh#1, 25%). Asterisks indicate fibers with myelin outfoldings. $P < 0.0001$, nonparametric one-way ANOVA, followed by Dunn's *post hoc* test, representative of two independent experiments. Western blot analysis of Nrg1 and Akt phosphorylation (S473) shows that activation of Akt decreases following Nrg1 type III downregulation (p-Akt/Akt/actin, ratio between *Mtmr2*$^{-/-}$ scramble-treated and *Mtmr2*$^{-/-}$ sh#1-treated using 12.5% LV is 1:0.85). Each lane is a pool of at least 10 different DRG/coverslips, representative of two independent experiments. Scale bar, 20 μm.

Data information: Results in (C–E) are expressed as mean ± SEM, Dunn's *post hoc* test, **$P < 0.01$; ***$P < 0.001$.
Source data are available online for this figure.

---

phosphorylation levels (Fig 4B and C), consistent with the role of Nrg1 type III in the control of myelin thickness.

**Niacin/Niaspan ameliorates hypermyelination in the Vimentin-null model, associated with increased Nrg1 type III expression**

As our proof-of-principle data suggested that downregulation of Nrg1 type III signaling ameliorates myelin outfoldings, particularly niacin (nicotinic acid), which is known to enhance Tace activity, we next performed *in vivo* studies using Niaspan, an extended release formulation of niacin, which is already used in clinical practice to lower cholesterol levels and increase HDL (high-density lipoprotein cholesterol) (Lukasova *et al*, 2011). To efficiently establish a Niaspan-based treatment protocol, we first considered the *Vim*$^{-/-}$ mouse model, in which hypermyelination *in vivo* and *in vitro* in the Schwann cell/DRG neuron co-culture system is the consequence of increased Nrg1 type III pathway activation (Triolo *et al*, 2012). Importantly, we previously reported that vimentin acts synergistically with Tace to negatively regulate myelination and that genetic reduction in Nrg1 type III in *Vim*$^{-/-}$ mice rescues hypermyelination (Triolo *et al*, 2012). First, we

confirmed that niacin treatment of *Vim*$^{-/-}$ co-culture explants rebalanced Akt activation and restored myelination (Fig 5A).

Next, we administered 160 mg/kg/day of Niaspan to *Vim*$^{-/-}$ mice by daily i.p. injection starting at P15 for 15 days. This dosage is intermediate within the range of 40–400 mg/kg, which has been already used in several preclinical trials (Chen *et al*, 2007, 2009; Zhang *et al*, 2008; Cui *et al*, 2010; Shehadah *et al*, 2010; Yan *et al*, 2012). We found that Niaspan enhanced Tace activity (Fig 5B) and rescued hypermyelination in *Vim*$^{-/-}$ nerves, as assessed by g-ratio analysis (Fig 5C). As hypermyelination in *Vim*$^{-/-}$ mice is due to increased Nrg1 type III signaling, this finding corroborates the hypothesis that activation of Tace and the consequent decrease in Nrg1 type III signaling represent an effective strategy to modulate myelination.

**Niaspan reduces myelin outfoldings in the nerve of *Mtmr2*$^{-/-}$ mice, a model of the CMT4B1 neuropathy**

We treated *Mtmr2*$^{-/-}$ mice using the same protocol as before, but daily for 2 months, as myelin outfoldings increase in number and progress in complexity with age (Bolino *et al*, 2004). We observed

---

**Figure 3.  Treatment of *Mtmr2*$^{-/-}$ Schwann cell/DRG neuron co-cultures using either rhTACE or niacin rescues myelin outfoldings.**

A    Titration of rhTACE on wild-type (WT) co-cultures stained for Mbp with quantification of Mbp-positive fibers and Schwann cell nuclei: $n = 18$ (NT, not treated), $n = 6$ (5 ng/ml), $n = 20$ (10 ng/ml), $n = 7$ (50 ng/ml) DRGs/coverslips from two independent experiments. $P = 0.0053$, nonparametric one-way ANOVA, followed by Dunn's *post hoc* test. Scale bar, 100 μm.

B    Immunohistochemistry and confocal microscopy of *Mtmr2*$^{-/-}$ cultures treated with 5 and 10 ng/ml rhTACE, with quantification of the percentage of myelin outfoldings; $n = 10$ (NT), $n = 10$ (5 ng/ml), $n = 11$ (10 ng/ml) DRGs/coverslips. A total of 547, 557, and 571 Mbp fibers were quantified, respectively. $P < 0.0001$, nonparametric one-way ANOVA, followed by Dunn's *post hoc* test. Western blot analysis of lysates from *Mtmr2*$^{-/-}$ cultures treated using 10 ng/ml rhTACE showing Nrg1 expression and Akt activation (S473 phosphorylation). Each lane is a lysate of at least $n = 10$ DRGs/coverslips. In *Mtmr2*$^{-/-}$ (10 ng/ml rhTACE) as compared to *Mtmr2*$^{-/-}$ (NT), Nrg1(150 kDa)/actin ratio is 0.8:1; Nrg1(65 kDa)/actin ratio is 0.48:1; and p-Akt/Akt/actin ratio is 0.6:1. Scale bar, 20 μm.

C    Titration of WT co-cultures using different amounts of niacin, nicotinic acid, with quantification of Mbp-positive fibers and Schwann cell nuclei, $n = 9$ (NT), $n = 8$ (1 mM), $n = 9$ (5 mM), $n = 8$ (10 mM) DRGs/coverslips. Representative of three independent experiments, $P = 0.0055$, nonparametric one-way ANOVA, followed by Dunn's *post hoc* test. Scale bar, 100 μm.

D    Immunohistochemistry and confocal microscopy of *Mtmr2*$^{-/-}$ cultures treated with 1 and 5 mM niacin, with quantification of the percentage of myelin outfoldings; $n = 31$ (NT), $n = 14$ (1 mM), and $n = 29$ (5 mM) DRGs/coverslips from two independent experiments. A total of 1,256, 489, and 1,305 fibers, respectively, were quantified. $P < 0.0001$, nonparametric one-way ANOVA, followed by Dunn's *post hoc* test. Tace activity was measured from lysates of niacin-treated cultures, $n = 7$ (KO NT) and $n = 8$ (KO 5 mM niacin) number of independent plates, each containing from 10 to 15 DRGs plated. Note that Tace activity was similar between WT and *Mtmr2*$^{-/-}$ cultures, one-tailed Mann–Whitney *U*-test, $P = 0.037$. Scale bar, 20 μm.

E    Niacin treatment of *Tace*$^{-/-}$ explants does not rescue hypermyelination, with quantification; $n = 8$ DRG/coverslips per condition; $P = 0.0056$, nonparametric one-way ANOVA, followed by Dunn's *post hoc* test. Representative of three independent experiments. Scale bar, 100 μm.

Data information: Results are expressed as mean ± SEM, Dunn's *post hoc* test, *$P < 0.05$; **$P < 0.01$; ***$P < 0.001$. Mbp, myelin basic protein; Nf, neurofilament.
Source data are available online for this figure.

that Niaspan significantly rescued myelin outfoldings in *Mtmr2*$^{-/-}$ nerves without altering myelin thickness (Fig 6A and B). Unfortunately, behavioral analysis and neurophysiology do not represent informative outcome measures at this age. Indeed, *Mtmr2*$^{-/-}$ mice display defects in the footprint gait analysis and a slowing of nerve conduction velocity at neurophysiological examination starting from 6 months of age (Bolino *et al*, 2004; Bolis *et al*, 2005). Bioclinical analyses performed to measure plasma or urine levels of ALT (alanine transaminase); ALP (alkaline phosphatase); AST (aspartate transaminase); HDL-C (high-density lipoprotein cholesterol); LDL (low-density lipoprotein cholesterol); TG (triglycerides); TBLI (bilirubin); DBLI (direct bilirubin, conjugated) confirmed that Niaspan administration

did not result in major side effects (Fig EV5). Interestingly, Tace has been recently found to promote OPCs differentiation in the CNS (Palazuelos *et al*, 2014). Thus, we analyzed CNS myelinated tracts in Niaspan-treated *Mtmr2*$^{-/-}$ mice and we found that CNS myelination was not altered (Fig EV6).

### Niaspan reduces tomacula in the nerve of Pmp22$^{+/-}$ mice, a model of the HNPP neuropathy

Next, we extended our strategy to the HNPP hypermyelinating neuropathy. HNPP is due to *PMP22* haploinsufficiency (Adlkofer *et al*, 1995, 1997) and is modeled by the *Pmp22*$^{+/-}$ mouse,

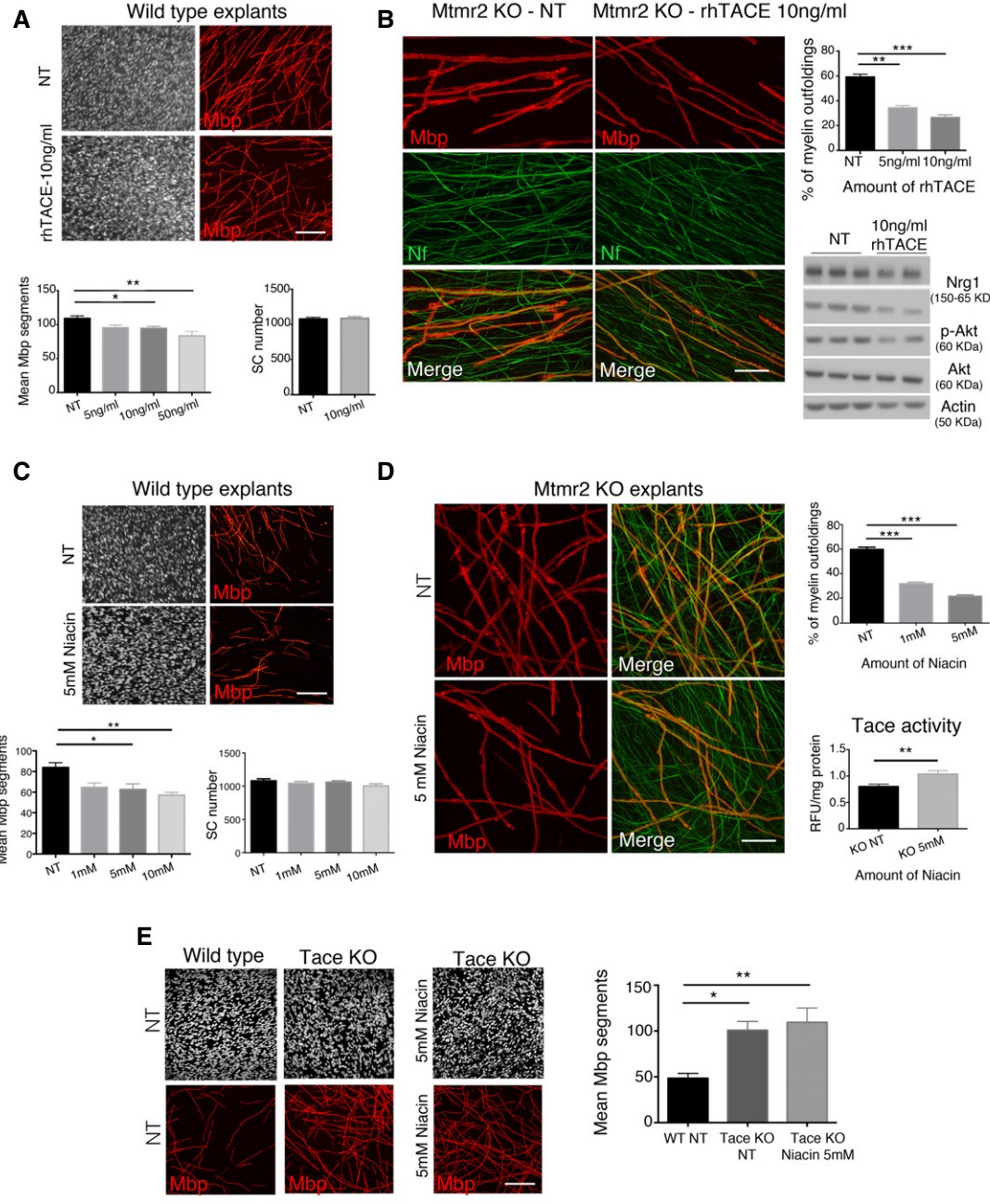

**Figure 3.**

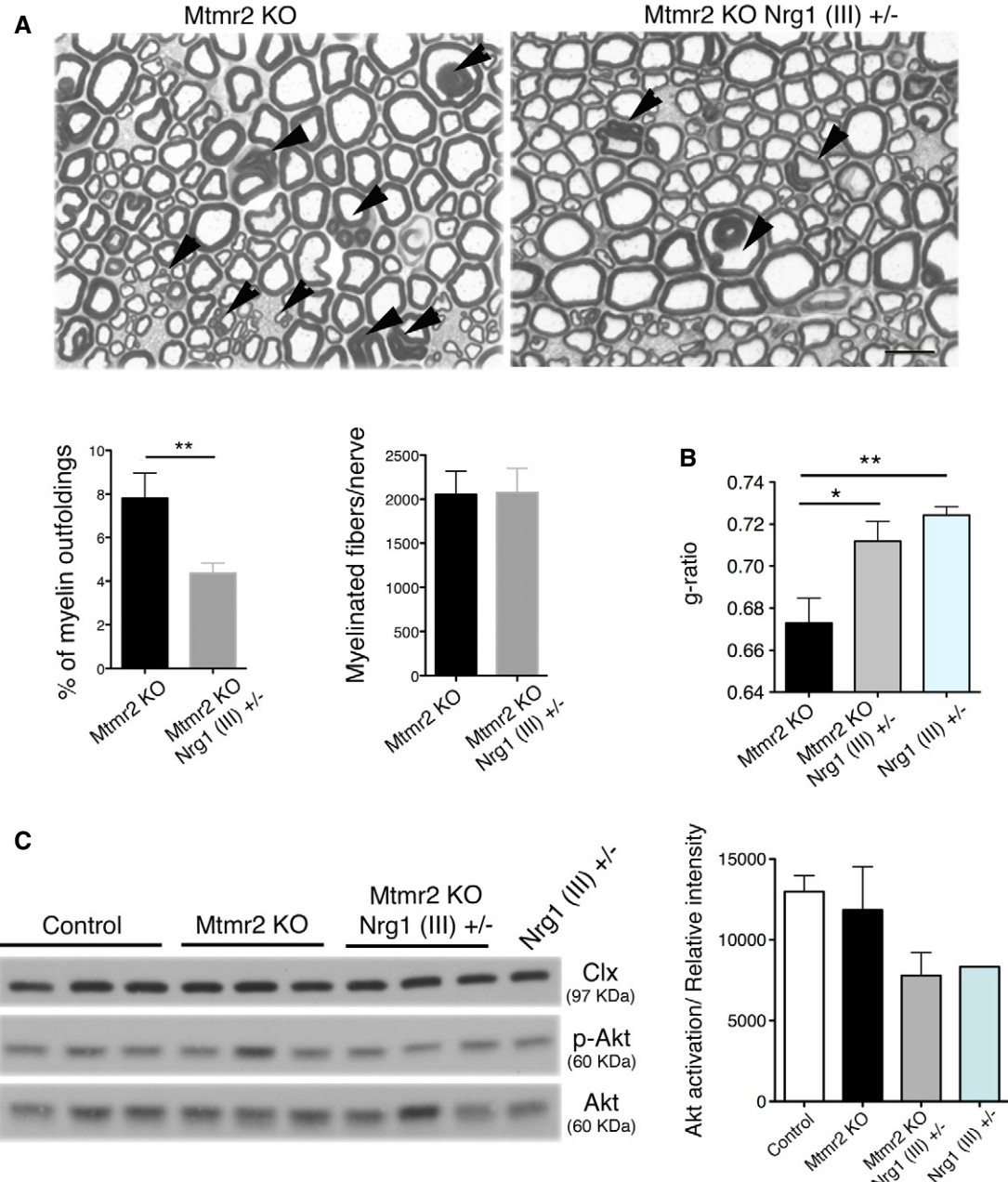

**Figure 4.**

whose nerves are characterized by tomacula (Adlkofer *et al*, 1995, 1997; Bai *et al*, 2010). We first confirmed that Akt phosphorylation is increased in $Pmp22^{+/-}$ nerves, in agreement with previous studies (Fig 7A; Fledrich *et al*, 2014). Importantly, we observed that Tace activity was reduced, whereas Tace protein levels were increased in $Pmp22^{+/-}$ nerves (Fig 7B), possibly indicating that these mice are physiologically trying to downregulate excessive myelin production. Notably, these findings also predicted that the Niaspan-mediated increase in Tace activity and the downregulation of the PI3K–Akt signaling would be effective in $Pmp22^{+/-}$ mice.

As predicted, we found that Niaspan rescued hypermyelination and reduced the number of tomacula in $Pmp22^{+/-}$ sciatic nerves

(Fig 7C–E). Unfortunately, no functional outcome measures are available for this model at the age of the analyses.

The $Pmp22^{-/-}$ mouse model displays a more severe neuropathy as compared to $Pmp22$ haploinsufficiency. In $Pmp22^{-/-}$ nerves, tomacula are abundant already at P24 and are associated with hypomyelination and fiber loss (Adlkofer *et al*, 1995) (Fig 8C). Later on, tomacula are more rare and nerves are characterized by chronic signs of demyelination such as onion bulbs (Adlkofer *et al*, 1995). These findings suggest that the complete absence of Pmp22 results in a severe neuropathy resembling Dejerine–Sottas disease, likely caused by a different pathogenetic mechanism as compared to haploinsufficiency. Accordingly, in $Pmp22^{-/-}$ sciatic nerves, we found decreased expression levels of Mbp and P0 myelin proteins

**Figure 4.  Genetic ablation of *Nrg1* type III in *Mtmr2*$^{-/-}$ mice rescues myelin outfoldings.**

A  Representative images of semithin section analysis of sciatic nerves from *Mtmr2*$^{-/-}$ and *Mtmr2*$^{-/-}$;*Nrg1* (III)$^{+/-}$ mice at 6 months. Myelin outfoldings (arrowheads) are reduced in *Mtmr2*$^{-/-}$;*Nrg1* (III)$^{+/-}$ nerves (n = 9) as compared to *Mtmr2*$^{-/-}$ (n = 8). Scale bar, 10 μm. \*\**P* = 0.0037, two-tailed Mann–Whitney *U*-test.

B  G-ratio analysis on sciatic nerves from *Mtmr2*$^{-/-}$, *Mtmr2*$^{-/-}$;*Nrg1* (III)$^{+/-}$, and *Nrg1* (III)$^{+/-}$ mice at 6 months indicates that myelin thickness of *Mtmr2*$^{-/-}$;*Nrg1* (III)$^{+/-}$ is similar to *Nrg1* (III)$^{+/-}$ nerves and different from *Mtmr2*$^{-/-}$, which is in the normal range as already reported (Bonneick *et al*, 2005), n = 5 animals per genotype. Mean g-ratios: *Mtmr2*$^{-/-}$ 0.67 ± 0.03, 2,129 fibers, n = 5 animals; *Mtmr2*$^{-/-}$;*Nrg1* (III)$^{+/-}$ 0.71 ± 0.03, 2,139 fibers, n = 5 animals; *Nrg1* (III)$^{+/-}$ 0.72 ± 0.03, 2,152 fibers, n = 4 animals. *Mtmr2*$^{-/-}$ as compared to *Mtmr2*$^{-/-}$;*Nrg1* (III)$^{+/-}$, \**P* = 0.0145; *Mtmr2*$^{-/-}$ as compared to *Nrg1* (III)$^{+/-}$, \*\**P* = 0.0030; and *Mtmr2*$^{-/-}$;*Nrg1* (III)$^{+/-}$ as compared to *Nrg1* (III)$^{+/-}$ *P* = 0.3193 (repeated-measures ANOVA).

C  Western blot analysis on lysates from *Mtmr2*$^{-/-}$, *Mtmr2*$^{-/-}$;*Nrg1* (III)$^{+/-}$, and *Nrg1* (III)$^{+/-}$ sciatic nerves at P10 indicates that Akt activation (S473 phosphorylation) is similar between *Mtmr2*$^{-/-}$;*Nrg1* (III)$^{+/-}$ and *Nrg1* (III)$^{+/-}$ consistent with g-ratio findings. Representative of three independent experiments. Phosphorylation of Akt is relative to total Akt normalized on calnexin (Clx) in the graph showing the quantification.

Data information: Results are expressed as mean ± SEM.
Source data are available online for this figure.

and normal Akt phosphorylation levels (Fig 8A and B). Contrary to what we observed in the *Pmp22*$^{+/-}$ model, Niaspan treatment did not reduce the number of tomacula, myelin thickness or increase the number of myelinated fibers in *Pmp22*$^{-/-}$ sciatic nerves (Fig 8C–E).

## Discussion

CMTs represent highly heterogeneous disorders caused by mutations in at least 70 different genes (Rossor *et al*, 2013). While much is known on the pathogenetic mechanisms, the high heterogeneity in their causes would suggest the development of a therapeutical approach specific for each subtype. Rather, the development of a common strategy aimed at restoring proper myelin thickness, preventing axonal loss, and favoring regeneration independently of the underlying pathogenetic mechanism might be the most valid approach.

Nrg1 type III is an essential instructive signal for PNS myelination and repair (Cohen *et al*, 1992; Carroll *et al*, 1997; Kwon *et al*, 1997; Fricker & Bennett, 2011). Nrg1 type III may also contribute to the pathogenesis of some CMT neuropathy (Gouttenoire *et al*, 2013; Fledrich *et al*, 2014). In fact, soluble administration of Nrg1 overcomes impaired nerve development in a CMT1A rat model, possibly by balancing PI3K–Akt and Mek–Erk signaling pathways (Fledrich *et al*, 2014). However, a therapeutic approach using soluble

recombinant human Nrg1, which stimulates ErbB2 receptors, might not be easily applied to humans due to possible side effects.

Here we suggest that modulation of Tace activity and thus of Nrg1 type III levels using niacin/Niaspan may represent an effective unifying therapeutical strategy to ameliorate demyelinating CMT neuropathies with focal hypermyelination. We previously showed that the α-secretase Tace cleavage of Nrg1 type III inhibits myelination as mutant mice lacking neuronal Tace are hypermyelinated and their phenotype remarkably resembles Nrg1 type III overexpressing mice (Michailov *et al*, 2004; La Marca *et al*, 2011). Although Fleck *et al* reported that specific Tace cleavage of Nrg1 may promote myelination (Fleck *et al*, 2013), our previous results showed that Tace knockout mice are hypermyelinated (La Marca *et al*, 2011), and our present data both *in vitro* and *in vivo* show that niacin/Niaspan-mediated enhancement of Tace activity is associated with reduced Nrg1 pathway activation and myelination.

Our data indicate that Niaspan reduces the number of myelin outfoldings and tomacula in CMT4B1 and HNPP models, respectively. CMT4B1 is a severe autosomal recessive demyelinating neuropathy characterized by childhood onset; muscular weakness and atrophy; sensory loss; severely decreased nerve conduction velocity, and redundant loops of myelin, called myelin outfoldings (Previtali *et al*, 2007). We first demonstrated that this neuropathy is caused by loss of the MTMR2 phospholipid phosphatase (Bolino *et al*, 2000; Hnia *et al*, 2012) and generated a faithful mouse model for the disease, the *Mtmr2*$^{-/-}$ mouse (Bolino *et al*, 2004). In

**Figure 5.  Niacin/Niaspan ameliorates hypermyelination in *Vim*$^{-/-}$ mouse nerves.**

A  Niacin treatment of *Vim*$^{-/-}$ explants rescues hypermyelination, n = 10 DRGs/coverslips per condition, with quantification, representative of three independent experiments, *P* = 0.0048, nonparametric one-way ANOVA, followed by Dunn's *post hoc* test (\*\**P* < 0.01). Western blot analysis on lysates from treated and not treated explants (at least 10 DRGs/coverslips per lane) shows that increased Akt activation (S473 phosphorylation) in *Vim*$^{-/-}$ explants is restored to normal levels following 5 mM niacin treatment. Scale bar, 50 μm.

B  Niaspan administration (daily i.p. injection of 160 mg/kg Niaspan starting at P15 for 15 days) enhances Tace activity in *Vim*$^{-/-}$ nerves at P30, n = 6 animals per genotype, \**P* = 0.0325, one-tailed Mann–Whitney *U*-test. Niaspan treatment of *Vim*$^{-/-}$ mice does not affect mouse growth, as the growth rate in WT (n = 8, saline; n = 6 Niaspan) and *Vim*$^{-/-}$ (n = 10 saline and n = 10 Niaspan) mice either saline- or Niaspan-treated is not significantly different, linear mixed-effects (LME) models.

C  Semithin section and g-ratio analyses of sciatic nerves at P30 show that Niaspan does not alter myelin thickness in WT nerves, whereas it restores myelin thickness to normal values in *Vim*$^{-/-}$ nerves. G-ratio values: WT saline, 0.71 ± 0.003, 1,872 fibers, n = 5 animals; WT Niaspan, 0.71 ± 0.004, 1,882 fibers, n = 5 animals; *Vim*$^{-/-}$ saline 0.69 ± 0.004, 2,313 fibers, n = 6 animals; *Vim*$^{-/-}$ Niaspan 0.72 ± 0.01, 1,616 fibers, n = 5 animals. WT saline as compared to *Vim*$^{-/-}$ saline, *P* = 0.0588; *Vim*$^{-/-}$ saline as compared to *Vim*$^{-/-}$ Niaspan, \**P* = 0.0431 (repeated-measures ANOVA). Representative of two independent experiments. The number of myelinated fibers is similar between the four groups as shown, *P* = 0.3042, nonparametric one-way ANOVA followed by Dunn's *post hoc* test. Scale bar, 10 μm for semithin sections (the four lateral panels) and 1 μm for ultrastructural analysis images (middle panels).

Data information: Results are expressed as mean ± SEM.
Source data are available online for this figure.

$Mtmr2^{-/-}$ nerves, the number of fibers containing myelin outfoldings and loops increases progressively with age as well as their complexity. However, the phenotype of this mutant is milder as compared to human CMT4B1. In $Mtmr2^{-/-}$ nerves at 6 months, nerve conduction velocity decreases of 6–8 m/s as compared to controls. Moreover, $Mtmr2^{-/-}$ mice and wild-type littermates show

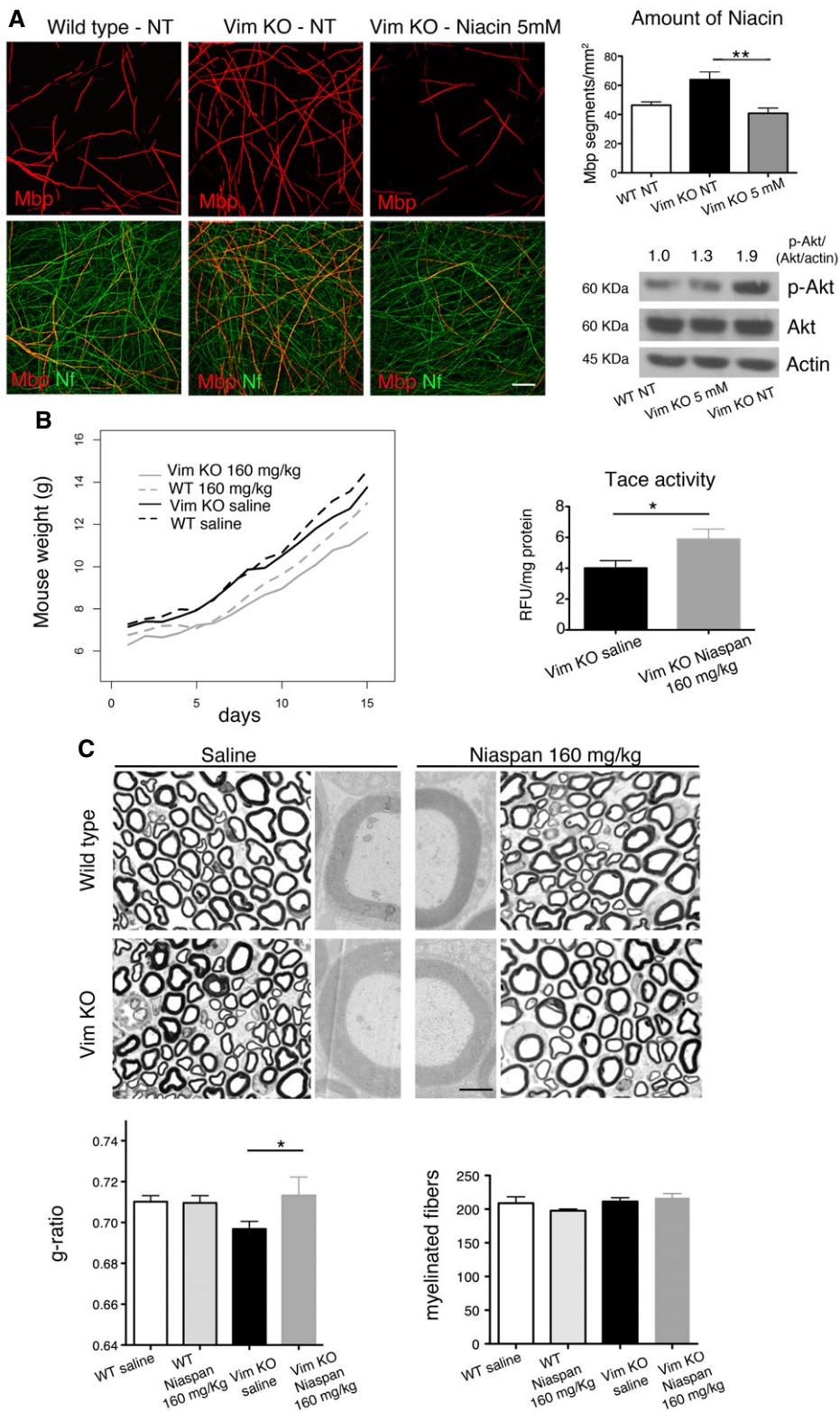

**Figure 5.**

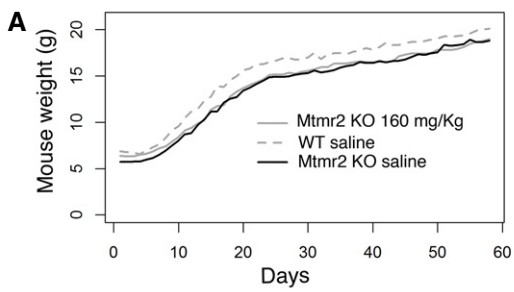

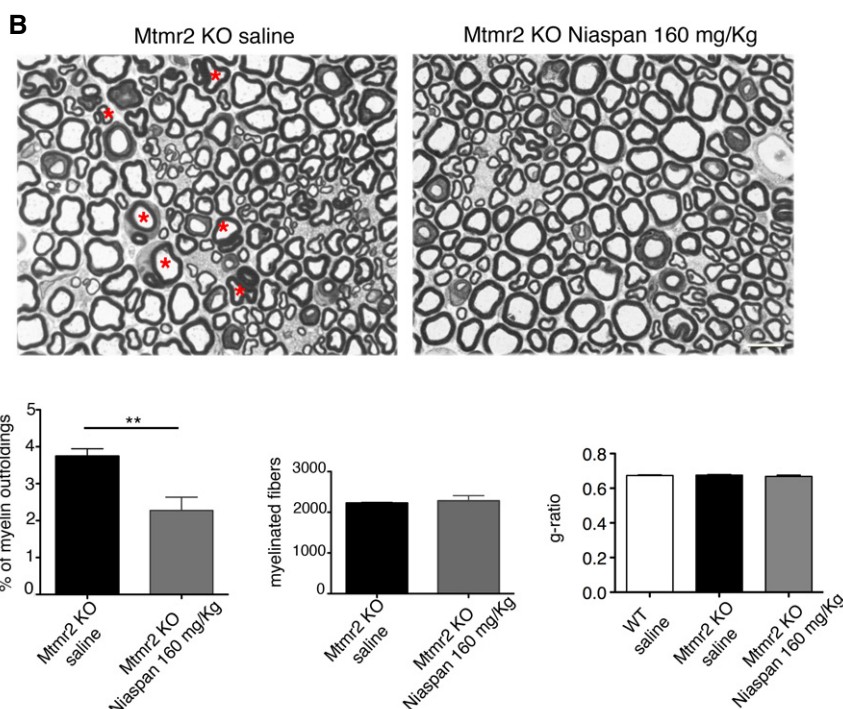

**Figure 6.  Niaspan reduces myelin outfoldings in the *Mtmr2⁻ᐟ⁻* mouse nerves.**

A  Niaspan administration (daily i.p. injection of 160 mg/kg starting at P15 for 60 days) does not affect the growth of *Mtmr2⁻ᐟ⁻* mice. The growth rates of *Mtmr2⁻ᐟ⁻* and WT saline-treated mice are significantly different, $P < 0.0001$, linear mixed-effects (LME) models, as already reported (Bolino *et al*, 2004). $n = 8$ animals per condition.

B  Niaspan administration reduces the percentage of myelin outfoldings (red asterisks) in *Mtmr2⁻ᐟ⁻* sciatic nerves without affecting the number of myelinated fibers as assessed by semithin section analysis at P75. Results are mean ± SEM, $n = 8$ mice per genotype; $**P = 0.0093$, two-tailed Mann–Whitney $U$-test; representative of three independent experiments. Niaspan treatment does not alter myelin thickness in WT or *Mtmr2⁻ᐟ⁻* mice as assessed by g-ratio analysis (WT saline-treated, 0.67 ± 0.003, 2,014 fibers; *Mtmr2⁻ᐟ⁻* Niaspan-treated, 0.67 ± 0.007, 1,924 fibers; *Mtmr2⁻ᐟ⁻* saline-treated, 0.68 ± 0.003, 2,244 fibers; $n = 4$ animals per condition). Scale bar, 10 μm.

---

no significant difference on rotarod testing, whereas gait analysis revealed only mild alterations in the mutants. Consistent with this, occasional degenerating axons have been noted in the nerves of these mutants only at 6 months. Thus, we could not perform behavioral analysis and neurophysiology to assess efficacy of Niaspan treatment at the functional level in *Mtmr2⁻ᐟ⁻* mice. However, we clearly showed a significant amelioration of the nerve pathology of Niaspan-treated mice and we can hypothesize that Niaspan treatment, by ameliorating the histological phenotype of CMT4B1, may also preserve axonal function/integrity. Of note, myelin outfoldings are predominant near, and probably arise from, juxtaparanodal/paranodal regions and represent unstable structures likely

perturbing axonal function (Bolino *et al*, 2004; Bonneick *et al*, 2005).

Even if tomacula are not unique to *PMP22* haploinsufficiency, they are the hallmark of HNPP (Adlkofer *et al*, 1995). The phenotype of the *Pmp22⁺ᐟ⁻* mouse, a model of human HNPP, is also mild (Adlkofer *et al*, 1997). In mutant nerves, the number of tomacula progressively increases in number with age. In myelinated fibers carrying tomacula, axonal displacement can be observed only at 10 months of age. Degeneration of tomacula (but not of axons) and demyelination have been reported in sciatic nerves of 15-month-old *Pmp22⁺ᐟ⁻* mice. Finally, neurophysiological examination revealed reduced M-amplitudes in mutant sciatic nerves only at 12- to

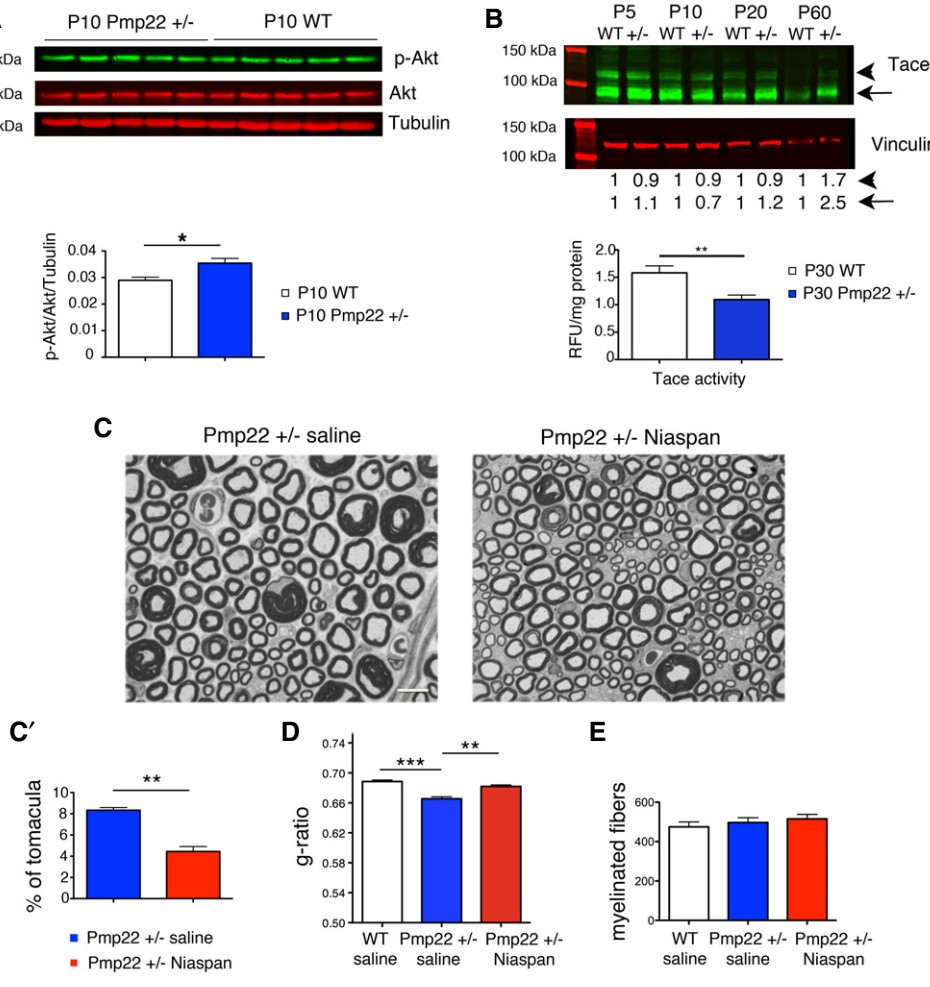

**Figure 7. Niaspan treatment ameliorates hypermyelination in *Pmp22*$^{+/-}$ mouse nerves.**

A   Western blot analysis of lysates from *Pmp22*$^{+/-}$ and WT sciatic nerves at P10 shows that Akt phosphorylation (S473) is increased in *Pmp22*$^{+/-}$ nerves, *$P$ = 0.0317, two-tailed Mann–Whitney $U$-test, representative of two independent experiments.

B   Western blot analysis of Tace using lysates from *Pmp22*$^{+/-}$ and WT sciatic nerves at different stages of development and below, Tace activity as measured in *Pmp22*$^{+/-}$ and WT sciatic nerves at P30 ($n$ = 6 mice per genotype, **$P$ = 0.0070, two-tailed Mann–Whitney $U$-test). The arrowhead indicates the pro-protein, and the arrow shows the cleaved active form of Tace. Representative of two independent experiments.

C   Representative images of semithin section analysis of sciatic nerves at P45 from *Pmp22*$^{+/-}$ saline and *Pmp22*$^{+/-}$ Niaspan-treated mice. Niaspan was administered daily by i.p. injection at 160 mg/kg starting at P15 for 30 days. Scale bar, 10 μm.

C′  The percentage of tomacula in sciatic nerves was assessed by ultrastructural analysis, $n$ = 6 animals per condition, **$P$ = 0.0022, two-tailed Mann–Whitney $U$-test, representative of two independent experiments.

D   G-ratio analysis on semithin sections of sciatic nerves shows that increased myelin thickness in *Pmp22*$^{+/-}$ sciatic nerves at P45 is restored to normal values following Niaspan treatment. Mean g-ratio values: WT saline-treated: 0.69 ± 0.002, 2,052 fibers, $n$ = 5 animals; *Pmp22*$^{+/-}$ saline-treated: 0.66 ± 0.003, 2,643 fibers, $n$ = 6 animals; *Pmp22*$^{+/-}$ Niaspan-treated, 0.68 ± 0.002, 2,311 fibers, $n$ = 6 animals. WT saline as compared to *Pmp22*$^{+/-}$ saline-treated, ***$P$ < 0.0001; *Pmp22*$^{+/-}$ saline-treated as compared to *Pmp22*$^{+/-}$ Niaspan-treated, **$P$ = 0.002; *Pmp22*$^{+/-}$ Niaspan-treated as compared to WT saline, $P$ = 0.0899 (repeated-measures ANOVA).

E   The number of myelinated fibers is similar between the three groups analyzed, $P$ = 0.4362, nonparametric one-way ANOVA, followed by Dunn's *post hoc* test. Numbers of animals analyzed are the same as in (D).

Data information: Results are expressed as mean ± SEM.
Source data are available online for this figure.

14-month-old mice. Thus, also for the *Pmp22*$^{+/-}$ model, neurophysiology and axonal degeneration cannot be used as outcome measures to assess efficacy of Niaspan treatment at least before 12–14 months of age. Interestingly, Niaspan ameliorates the phenotype of *Pmp22*$^{+/-}$ nerves with tomacula and increased levels of Akt phosphorylation, but not of *Pmp22*$^{-/-}$ mutant, which is instead primarily characterized by severe hypomyelination and axonal loss,

along with tomacula and normal levels of Akt phosphorylation. Thus, we may speculate that the complete absence of Pmp22 interferes with the trafficking and/or assembly of other myelin proteins and lipids, suggesting a different pathogenetic mechanism in the *Pmp22*$^{-/-}$ as compared to haploinsufficiency.

Niaspan-mediated amelioration of tomacula in *Pmp22*$^{+/-}$ nerves might be important to preserve axonal integrity and function. In

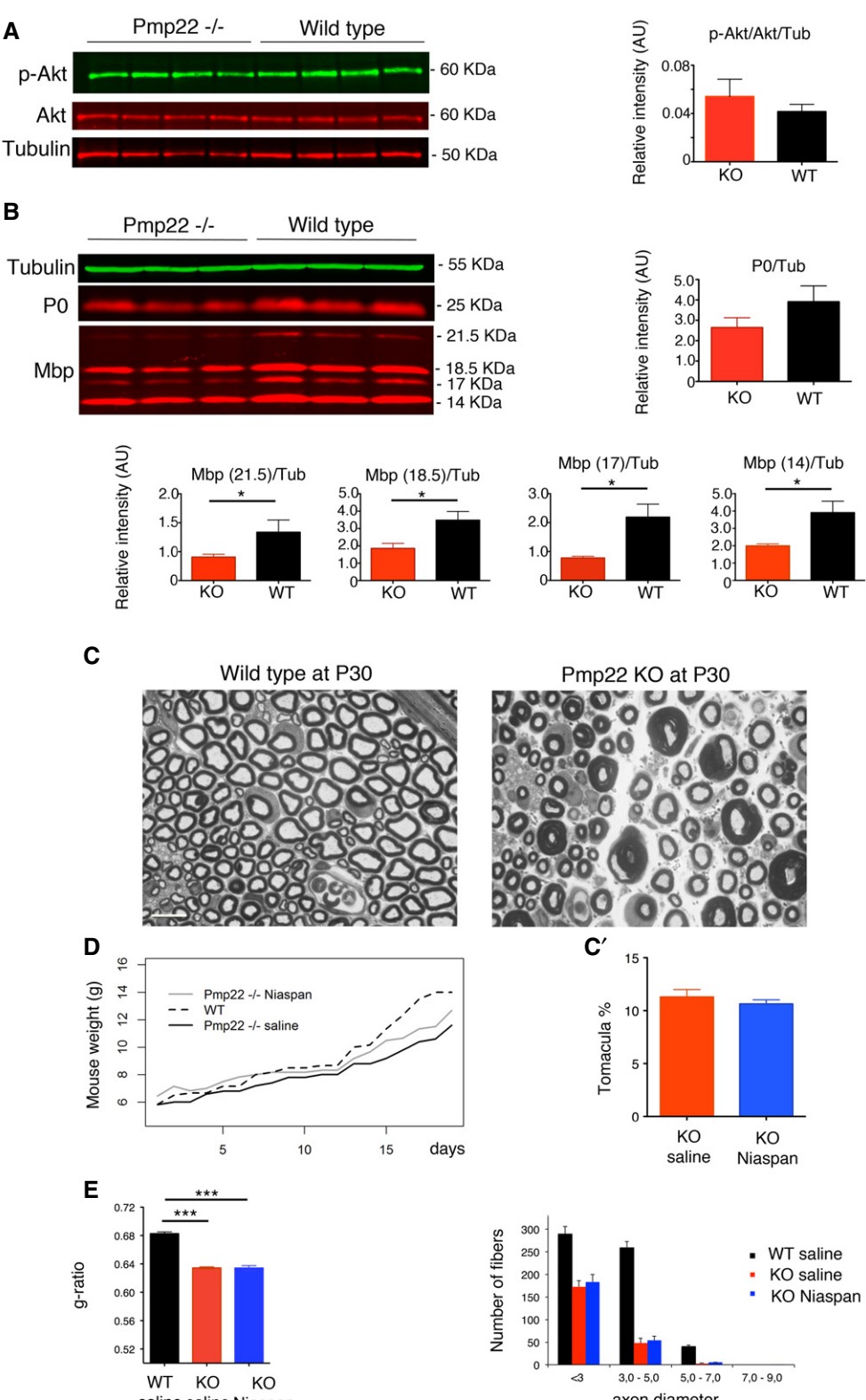

**Figure 8.**

fact, tomacula are unstable and uncompacted structures which tend to degenerate likely leading to axonal dysfunction (Adlkofer *et al*, 1995, 1997; Li *et al*, 2013). This is supported by the following

observations: (i) in older *Pmp22* mutants, reduction in tomacula parallels the increase in demyelination (Adlkofer *et al*, 1997); (ii) tomacula form first and then induce focal axon constriction, thus

**Figure 8.  Niaspan treatment does not ameliorate hypermyelination in _Pmp22_$^{-/-}$ mouse nerves.**

A   Western blot analysis to detect phosphorylation levels of Akt (S473) on lysates from _Pmp22_$^{-/-}$ and WT mice at P20, with quantification. $P = 0.6857$, two-tailed Mann–Whitney _U_-test, representative of three independent experiments.

B   Western blot analysis for Mbp and P0 on lysates from _Pmp22_$^{-/-}$ and WT mice P20 shows decreased myelin protein expression levels at both time points in _Pmp22_$^{-/-}$ nerves, *$P < 0.05$ for Mbp, but not for the P0 protein level at P20, for which the difference between _Pmp22_$^{-/-}$ and WT nerves is not statistically significant, $P = 0.1$; one-tailed Mann–Whitney _U_-test; representative of two independent experiments.

C   Representative images of semithin section analysis of WT and _Pmp22_$^{-/-}$ sciatic nerves at P30. Tomacula are abundant in _Pmp22_$^{-/-}$ nerves at this age, where both fibers with thicker myelin (in the range of < 3–4 μm in diameter) and thinner myelin (with diameters greater than 4 μm, particularly in motor fascicles) can be observed. Scale bar, 10 μm.

C′  Niaspan does not ameliorate the phenotype in _Pmp22_$^{-/-}$ mice, as indicated by the number of tomacula (quantification of the percentage of tomacula in the entire sciatic nerve section, $P = 0.7400$, Mann–Whitney _U_-test, $n = 15$ animals per condition).

D   Growth curve of _Pmp22_$^{-/-}$ mice treated with Niaspan administered by i.p. injection daily at 160 mg/kg and starting at P15 for 15 days. A significant time by group effect is noted between WT saline and _Pmp22_$^{-/-}$ saline (group effect $P = 0.9907$ and time effect $P = 0.0057$), indicating that the two groups start with similar weight values but then the WT saline group ($n = 6$ animals) grows more in time. When _Pmp22_$^{-/-}$ saline ($n = 5$) and _Pmp22_$^{-/-}$ Niaspan-treated ($n = 6$) are compared, the group effect is statistically significant ($P = 0.0076$) while the time effect is not ($P = 0.6677$), suggesting that these two groups have a similar growth trend but _Pmp22_$^{-/-}$ Niaspan-treated group starts with a higher baseline weight.

E   Niaspan does not ameliorate myelin thickness as indicated by g-ratio analysis, with axonal diameter distribution: WT $0.68 \pm 0.003$, 2,945 fibers, $n = 5$ animals; _Pmp22_$^{-/-}$ $0.64 \pm 0.003$, 1,557 fibers, $n = 6$ animals; _Pmp22_$^{-/-}$ Niaspan $0.63 \pm 0.003$, 1,445 fibers, $n = 6$ animals. WT saline as compared to _Pmp22_$^{-/-}$ saline-treated ***$P < 0.0001$; WT saline as compared to _Pmp22_$^{-/-}$ Niaspan-treated ***$P < 0.0001$ (repeated-measures ANOVA). Note the loss of myelinated fibers in _Pmp22_$^{-/-}$ nerves for all caliber axons.

Data information: Results in (A, B, C', and E) are expressed as mean $\pm$ SEM.
Source data are available online for this figure.

increasing the resistance, as demonstrated in the _Mag_$^{-/-}$ model, which is also characterized by tomacula formation in the nerve (Li _et al_, 2013); and (iii) tomacula preferentially arise at nodal–paranodal regions, thus also reducing the axon insulation (Adlkofer _et al_, 1995, 1997). All these events are thought to predispose _Pmp22_ mutant nerves to conduction blocks following compression, a neurophysiological feature of HNPP (Bai _et al_, 2010). A more recent hypothesis implies that Pmp22 deficiency may also affect the assembly of junction protein complexes during development and alter myelin permeability, thus predisposing to conduction blocks (Guo _et al_, 2014). However, it is unclear how much the increase in myelin permeability in Pmp22 deficiency contributes to conduction blocks as compared to tomacula-mediated axon constriction and myelin uncompaction.

We propose here that niacin/Niaspan may ameliorate hypermyelinating neuropathies independently of the underlying molecular defect. This strategy might also be effective in other CMTs characterized by myelin outfoldings or tomacula such as CMT4B2, CMT4B3, and CMT4H (Previtali _et al_, 2007; Nakhro _et al_, 2013). However, both myelin outfoldings and tomacula have been linked to dysregulation of the PI3K–Akt–mTOR pathway downstream of Nrg1 type III-ErbB2 receptors.

A mouse mutant with constitutive Akt activation specifically in Schwann cells displays enhanced myelination and mostly tomacula in peripheral nerves (Domenech-Estevez _et al_, 2016). Moreover, enhanced Akt phosphorylation levels have been detected in _Pmp22_$^{+/-}$ nerves with tomacula (Fig 7A; Fledrich _et al_, 2014). Tomacula have been also observed in _Pten_$^{-/-}$ nerves, in which elevated levels of the PtdIns(3,4,5)$P_3$ (also known as PIP$_3$) phospholipid lead to Akt–mTOR overactivation (Goebbels _et al_, 2012). Thus, it can be speculated that tomacula may derive from over-activation of the Akt–mTOR pathway.

We recently suggested that in _Mtmr2_$^{-/-}$ nerves increased levels of PtdIns(3,5)$P_2$ are at the basis of myelin outfolding formation (Vaccari _et al_, 2011). Interestingly, PtdIns(3,5)$P_2$, a key regulator of membrane trafficking at the level of late endosomes and lysosomes

(Di Paolo & De Camilli, 2006), can be also generated from PIP$_3$. Consistent with this, myelin outfoldings have also been reported in the _Pten_$^{-/-}$ mutant nerves with enhanced PIP$_3$ levels (Goebbels _et al_, 2010), suggesting that myelin outfoldings may represent a PIP$_3$-related phenomenon (Goebbels _et al_, 2012). The observation that rapamycin, which is a mTORC1 inhibitor, reduces tomacula more efficiently than myelin outfoldings in the _Pten_ mutant may support the conclusion that myelin outfoldings might be related to phospholipid levels dysregulation, whereas tomacula to enhanced Akt–mTOR pathway activation (Goebbels _et al_, 2012). Of note, rapamycin has major side effects and is not suitable for long-term continuous treatment of chronic disorders such as peripheral neuropathies.

Niaspan has been extensively used in clinical practice in humans to increase HDL (high-density lipoprotein cholesterol), decrease cholesterol levels, and reduce the mortality of cardiovascular events (Lukasova _et al_, 2011). More recently, it has been shown that this drug has additional functions, depending on the cell-specific mechanism of action and the dosage (Lukasova _et al_, 2011; Offermanns & Schwaninger, 2015). Among the lipid-independent effects, Niaspan is also thought to increase synaptic plasticity either directly, by promoting BDNF (brain-derived neurotrophic factor) and TrkB (tyrosine receptor kinase) expression and function in neurons, or indirectly through HDL (Chen _et al_, 2014), suggesting a potential beneficial effect in regeneration and repair, which is a relevant clinical aspect in CMT.

## Materials and Methods

### Animals

All experiments involving animals were performed in accordance with Italian national regulations and covered by experimental protocols reviewed by local Institutional Animal Care and Use Committees (IACUC 701 and 664).

The generation and genotyping of $Mtmr2^{-/-}$, $Vim^{-/-}$, $Tace^{-/-}$, $Nrg1$ (III)$^{+/-}$, $Pmp22^{+/-}$, and $Pmp22^{-/-}$ mouse mutants have been already described (Adlkofer *et al*, 1995; Bolino *et al*, 2004; La Marca *et al*, 2011; Triolo *et al*, 2012).

For PCR, we isolated DNA from tail biopsies using DirectPCR lysis reagent (Viagen Biotech), following manufacturer's directions.

Animals were randomly included into experimental groups according to genotyping, age, and sex. No animals had to be excluded due to illness in all the experiment performed. Animal experiments (morphological analyses) were performed in a blinded fashion toward the investigator. Investigators performing animal handling, sampling, euthanasia, and raw data analysis were not blinded.

**Morphological analysis**

Semithin analysis of sciatic nerves and ultrastructural analysis of sciatic and optic nerves were performed as described previously (Wrabetz *et al*, 2000).

To perform morphometric analysis, digitalized images of fiber cross sections were obtained from corresponding levels of the quadriceps or sciatic nerves with a 100× objective and Leica DFC300F digital camera (Milan, Italy). Five images per animal were analyzed using the Leica QWin software (Leica Microsystem) and the g-ratio calculated as the ratio between the mean diameter of an axon (without myelin) and the mean diameter of the same axon including the myelin sheath.

For morphometric analysis on ultrastructural sections, 20 images per animal were taken at 4,000× (LEO 912AB Transmission Electron Microscope, Milan, Italy) and the g-ratio values determined by measuring axon and fiber diameters.

**Primary cell culture**

*Schwann cell/DRG neuron co-cultures*
Myelin-forming Schwann cell/DRG neuron co-cultures were established from E13.5 mouse embryos as previously described (Bolis *et al*, 2009). For myelination, DRGs were placed on C-media supplemented with ascorbic acid for 7–15 days (50 μg/ml, SIGMA).

Co-cultures were treated using niacin (SIGMA) diluted in MEM and with rhTACE reconstituted in sterile water (R&D Systems).

*Purified neuronal culture*
Purified rat DRG neurons were established as described, but DRGs were first incubated with trypsin (0.25%) for 45 min at 37°C. Cells were also mechanically dissociated and then plated at a concentration of one to two DRGs per glass coverslip. Cells were subjected to three cycles of antimitotics (fluorodeoxyuridine and uridine, Sigma) in NB medium to remove fibroblasts and Schwann cells.

*Purified rat Schwann cell culture*
Isolated rat Schwann cells were prepared as reported previously (Taveggia *et al*, 2005) and cultured using DMEM with 10% of fetal calf serum, 2 ng/ml recombinant human Neuregulin 1-β1 (R&D Systems), and 2 mM forskolin (Calbiochem).

**Lentivirus preparation (LV) and infection**

To downregulate *Nrg1* (III) expression *in vitro*, non-concentrated lentiviral vectors (LV) carrying shRNA for *Nrg1* (III) were used to transduce rat purified neuronal cultures and mouse explants (Thermo Scientific, clone ID #TRCN0000068234 and #TRCN0000068236, pKLO1 vector). *Tace* expression in $Mtmr2^{-/-}$ cultures was downregulated using shRNA LVs as reported in La Marca *et al* (2011) (Thermo Scientific, clone ID TRCN0000031949, TRCN0000031952, and TRCN0000031953). Non-concentrated LVs were produced as already reported (Bolis *et al*, 2009). Quantitative RT–PCR to assess *Tace* expression downregulation in isolated Schwann cells transduced with Tace shRNA LVs was performed as previously described (Bolis *et al*, 2005; La Marca *et al*, 2011).

To preferentially target neurons in $Mtmr2^{-/-}$ or wild-type explants using LVs, infection was carried out the day after plating the DRGs for 24 h using C-media, thus before Schwann cell migration along axons.

**Tace activity measurements**

For Tace activity determination, the SensoLyte® 520 TACE (α-Secretase) Activity Assay Kit *Fluorimetric* was used (Anaspec). The SensoLyte® 520 TACE Activity Assay Kit contains a QXL™520/5-FAM FRET substrate, derived from a sequence surrounding the cleavage site of TACE. In the intact FRET peptide, the fluorescence of 5-FAM is quenched by QXL™520. Active TACE cleaves FRET substrate into two separate fragments resulting in an increase in 5-FAM fluorescence which can be monitored at excitation/emission = 490 nm/520 nm. The long wavelength fluorescence of 5-FAM is less interfered by the autofluorescence of cell components and test compounds. Lysates from sciatic nerves (single nerves at P30) or DRG explants (pools of 10–15 DRGs) were prepared using an assay buffer containing 0.1% Triton X-100. Samples were sonicated, kept at 4°C for 30 min on a rotating wheel and then spin at 20,000 × $g$ at 4°C for 15 min. Supernatant was collected, and protein determination was performed using BCA assay (Pierce, Thermo Scientific). The preparation of Tace substrate, the standard curve, and the positive control provided by the assay were performed following manufacturer's conditions. Tace substrate was diluted 1:100 in the assay buffer. The standard 5-FAM was diluted from the 1 mM stock to a final concentration of 4 μM, and a standard curve with a range from 2 to 0.03 μM in assay buffer (seven points) was made. Tace substrate was added to the blank, the positive control (rhTACE, recombinant human TACE) and to the sample lysates. When ready, the 96-well plate was gently mixed for 10 min and then incubated at RT for 50 min. For each sample, using a fluorescence microplate reader detecting emission at 520 nm with excitation at 490 nm (Victor³, PerkinElmer), the fluorescence intensity was normalized for the protein content and the same wild-type sample was used as a calibrator to compare independent experiments.

**Antibodies**

The following primary antibodies were used: rabbit anti-Neuregulin 1α/β1/2 (C20) (sc-348; Santa Cruz Biotechnology); rabbit anti-phospho-Akt (Ser473) (D9W) (4060; Cell Signaling); rabbit anti-Akt (pan) (C67W7) (4691; Cell Signaling); rabbit anti-actin (A2066;

Sigma-Aldrich); rat anti-myelin basic protein on mouse explants (hybridoma, kindly provided by Dr. Virginia Lee); chicken anti-neurofilament NF-M (PCK-593P; Covance); mouse anti-β-tubulin (T4026; Sigma-Aldrich); rabbit anti-Tace (AB39162; Abcam); mouse anti-vinculin (V284) (05-386; Millipore); rabbit anti-calnexin (C4731; Sigma-Aldrich); rabbit anti-phospho-p44/42 MAPK (Erk1/2) (Thr202/Tyr204) (9101; Cell Signaling); rabbit anti-p44/42 MAP kinase (9102; Cell Signaling); rabbit anti-p-Neu (Tyr1248)-R (i.e., p-ErbB-2) (sc-12352-R; Santa Cruz Biotechnology); rabbit anti-Neu (C-18) (i.e., ErbB-2) (sc-284; Santa Cruz Biotechnology); rabbit anti-NF-L (C28E10) (2837; Cell Signaling); chicken anti-myelin protein zero (P0) (AB9352; Millipore); rat anti-myelin basic protein (MAB386; Millipore).

The rabbit anti-Neuregulin 1α/β1/2 (C20, Santa Cruz Biotechnology) antibody recognizes all Nrg1 isoforms with an "a" tail. The 150 kDa band is specific for Nrg1 type III, while the 65 kDa identifies the cleaved Nrg1 form, which could belong to several Nrg1 isoforms.

For immunofluorescence, secondary antibodies included fluorescein (FITC)-conjugated (715-095-151; 711-095-152; 712-095-153) and rhodamine (TRITC)-conjugated (715-025-150; 711-025-152; 712-025-150) donkey anti-mouse or rabbit or rat IgG (Jackson ImmunoResearch). For Western blotting, secondary antibodies included horseradish peroxidase (HRP)-conjugated goat anti-rabbit, rabbit anti-mouse and rabbit anti-rat IgG (Dako), and IRDye 800- (926-32210; 926-32219) and 680-conjugated (926-68071; 926-68028) goat anti-mouse, goat anti-rabbit, goat anti-rat, and donkey anti-chicken IgG (Li-Cor Biosciences).

### Immunohistochemistry and analysis of myelination

Schwann cell/DRG neuron co-cultures were fixed for 15 min in 4% paraformaldehyde, permeabilized for 5 min in ice-cold methanol at −20°C, blocked for 20 min with 10% NGS, 1% BSA, and then incubated with primary antibody for 1 h. After washing, the coverslips were incubated with the secondary antibody for 30 min, washed, and mounted. For double immunostaining with anti-Nf-l and anti-Mbp antibodies, the coverslips were blocked with 1% BSA, 10% NGS for 20 min and primary antibodies were incubated overnight at 4°C.

To quantify the amount of myelin, using a fluorescence microscope at least five fields/coverslip were randomly acquired and Mbp-positive myelinated fibers were counted per field. Means of each coverslip/DRG have been used as different "*n*" for statistical analysis.

To quantify myelin outfoldings, at least 300 Mbp-positive myelinated fibers were evaluated, from "*n*" different DRG explants/coverslips. The percentage of Mbp-positive fibers showing myelin outfoldings among the total number of Mbp-positive fibers was indicated.

### Western blot analysis

Protein lysates from mouse sciatic nerves for Western blot analysis were prepared using a lysis buffer containing 2% SDS, 50 mM Tris buffer pH 8.0, 150 mM NaCl, 10 mM NaF, 1 mM NaVO₃, complete protease and phosphatase inhibitors (Roche). To prepare lysates from co-cultures, a lysis buffer containing 1% Triton X-100 was used and pools of at least 10–12 coverslips/DRGs for each

determination were prepared. Protein quantification was performed using BCA assay (Pierce, Thermo Scientific).

SDS–PAGE gels were performed as already reported (Bolis *et al*, 2009). Immunoblots were revealed by using either ECL/ECL-prime developing systems and films for chemiluminescent detection (Amersham) or by Odyssey CLx Infrared Imaging System (Li-Cor Biosciences).

### Statistical analyses

Power analyses were performed using GPower software, v. 3.1 (Faul *et al*, 2009), based on a two independent sample Mann–Whitney *U*-test (one-tailed) with a significance level α set equal to 5%. Power analyses were performed after conducting experiments (*a posteriori*).

For each analysis, we evaluated whether the assumption required for correct application of standard parametric tests were met. Thus, two-tailed nonparametric Mann–Whitney *U*-test was performed to compare two independent groups. Only in the case of TACE activity measured as a consequence of niacin/Niaspan treatment (specific *a priori* direction hypothesis), a one-tailed Mann–Whitney *U*-test was applied as the drug is known to increase TACE enzyme activity.

One-sample Wilcoxon test was performed to assess whether the median of ratios was greater than one (one-tailed test). The exact *P*-value was calculated.

In presence of more than two groups, Kruskal–Wallis test, the nonparametric one-way ANOVA counterpart, was applied followed by Dunn's *post hoc* correction.

G-ratio analysis was performed by applying repeated-measures ANOVA (linear mixed-effects model (LME) framework) (Laird & Ware, 1982), calculated to properly account for dependency structure induced by the experiment (same nerve section measured multiple times using five random images for each mouse).

Growth curve data were modeled by applying linear mixed-effects (LME) models. In the growth curve model, we included as covariates the treatment group and the time variable along with their interaction to highlight potential differences in the growth dynamics. Mixed-effect models are flexible models that allow to include in the model additional random-effect terms: Random intercept and random slopes were specified in our model meaning that each mouse can have its own longitudinal trajectory. The inclusion of random components is fundamental to account for unobserved biological heterogeneity. When appropriate, a logarithmic transformation was applied to linearize the trend.

LME models were fitted in R (version 3.1.2) by using the *nlme* package.

In evaluating statistical significance, a 5% level was used in the analyses.

**Expanded View** for this article is available online.

### Acknowledgements

We are grateful to Ueli Suter for providing the *Pmp22*-deficient mice and Rosa La Marca for technical contribution. We thank the Hematologic testing Laboratory, Ospedale San Raffaele Mouse Clinic, for bioclinical analysis. Part of this work was carried out in the Alembic (Advanced Light and Electron Microscopy BioImaging Center) facility of the Ospedale San Raffaele, Milan,

### The paper explained

**Problem**

Charcot–Marie–Tooth (CMT) neuropathies have a collective prevalence of 1:2,500 and as a whole represent the most common form of human hereditary neuromuscular disease. CMTs are due to mutations in at least 70 different genes and are commonly characterized by distal wasting, weakness, and sensory loss (Rossor *et al*, 2013). No effective treatments are known for any CMT subtype. Although clinical and pathological features partially overlap, the molecular mechanisms at the basis of CMTs are highly heterogeneous. Thus, it is difficult to envisage a single suitable treatment for all pathogenetic mechanisms. Further, any designed therapy, besides correcting the genetic/metabolic defect should promote nerve regeneration and remyelination, which is the main cause of morbidity.

**Results**

Axonal Neuregulin 1 (Nrg1) type III is a key growth factor controlling the amount of PNS myelin and has been recently implicated in regeneration (Stassart *et al*, 2013). We previously showed that the α-secretase Tace inhibits Nrg1 activity and hence PNS myelination (La Marca *et al*, 2011). Interestingly, niacin/Niaspan (nicotinic acid) is a drug known to enhance Tace activity. Thus, we postulated that Niaspan, by enhancing Tace activity and modulating Nrg1 type III, could benefit hypermyelinating neuropathies characterized by excessive focal myelin. Our data indicate that Niaspan treatment reduces the number of myelin outfoldings in the $Mtmr2^{-/-}$ mouse, a model of CMT type 4B1 neuropathy. We also report here that Niaspan is effective in reducing the number of tomacula in the $Pmp22^{+/-}$ mouse, a model of the HNPP (hereditary neuropathy with liability to pressure palsies) neuropathy.

**Impact**

Here, we provide evidence that Niaspan—by modulating Tace activity and, hence, PNS myelination—represents a valid approach for the treatment of CMT4B1 and HNPP neuropathies, which may be extended to other forms of CMT characterized by excessive myelin such as CMT4B2, B3, and CMT4H.

Niaspan is a FDA-approved drug, which has been extensively used to decrease lipid levels and prevent atherosclerosis (in humans in gram dose ranges). Depending on the dose and the cell-specific mechanism of action, Niaspan has also lipid-independent effects. For example, it is known to protect neuronal function, promote synaptic plasticity, and reduce inflammation, suggesting a potential beneficial effect in regeneration and repair, which is a relevant clinical aspect in CMT (Lukasova *et al*, 2011).

Italy. A.B. was supported by Telethon-Italy (GPP10007D, GGP12017, and GGP15012A); Association Française contre les Myopathies (AFM)-France (#16040 and 16922), and the ERA-Net for Research Programmes on Rare Diseases (2015 CMT-NRG); C.T. by Telethon-Italy (GPP10007 and GGP15012); S.C.P. by Telethon-Italy (GGP12024, GPP10007B, GGP15012B), by AFM-France (#15-18518) and the Italian Ministry of Health (RF-2011-02347127); MG-V by the San Raffaele International Postdoctoral Programme PCOFUND-GA-2010-267264 INVEST.

### Author contributions

AB and SCP conceived and designed the experiments. FP, VA, MG-V, MP, RN, and CR performed the experiments. AB, FP, CT, MG-V, MP, PD, CB, AN, and SCP analyzed the data. AB wrote the manuscript.

### Conflict of interest

The authors declare that they have no conflict of interest.

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
