## [Review Process File · EMBO Molecular Medicine]

Niacin-mediated Tace activation ameliorates CMT neuropathies with focal hypermyelination

Alessandra Bolino, Françoise Piguet, Valeria Alberizzi, Marta Pellegatta, Cristina Rivellini, Marta Guerrero-Valero, Roberta Nosedà, Chiara Brombin, Alessandro Nonis, Patrizia D'adamo, Carla Taveggia, Stefano Previtali

Corresponding author: Alessandra Bolino, San Raffaele Scientific Institute

Review timeline:

Submission date:	27 February 2016
Editorial Decision:	20 March 2016
Revision received:	26 August 2016
Editorial Decision:	15 September 2016
Revision received:	22 September 2016
Accepted:	04 October 2016

Transaction Report:

Editor: Roberto Buccione

1st Editorial Decision

20 March 2016

Thank you for the submission of your manuscript to EMBO Molecular Medicine. We have now heard back from the three Reviewers whom we asked to evaluate your manuscript.

Although all three reviewers agree on the potential importance and interest of your work, they do find a number of shortcomings that preclude publication in its current form.

I will not dwell into much detail, but I would like to highlight the main items of concern: 1) the need for behavioural and/or electrophysiological studies or at the least analysis of axonal degeneration and 2) lack of firm demonstration that indeed Niaspan is acting through TACE as suggested. There are of course other important items of concern/requests.

During our reviewer cross-commenting, the reviewers agreed that analysis of behavior/electrophysiology/axon degeneration should be attempted or its absence should be discussed much more thoroughly and convincingly. I would invite you to provide further experimentation in this respect in addition of course to addressing the mechanism of action of Niaspan.

We are thus prepared to consider a revised submission, with the understanding that the above two main Reviewer concerns (in addition to the points) must be addressed with additional experimental data where appropriate and that acceptance of the manuscript will entail a second round of review. The overall aim is to significantly upgrade the clinical relevance and conclusiveness of the dataset, which of course is of paramount importance for our title.

Please note that it is EMBO Molecular Medicine policy to allow a single round of revision only and that, therefore, acceptance or rejection of the manuscript will depend on the completeness of your responses included in the next, final version of the manuscript.

EMBO Molecular Medicine now requires a complete author checklist (<http://embomolmed.embopress.org/authorguide#editorial3>) to be submitted with all revised manuscripts. Provision of the author checklist is mandatory at revision stage; The checklist is designed to enhance and standardize reporting of key information in research papers and to support reanalysis and repetition of experiments by the community. The list covers key information for figure panels and captions and focuses on statistics, the reporting of reagents, animal models and human subject-derived data, as well as guidance to optimise data accessibility. This checklist especially relevant in this case given the issues raised with respect to statistical treatment and animal numbers.

As you know, EMBO Molecular Medicine has a "scooping protection" policy, whereby similar findings that are published by others during review or revision are not a criterion for rejection. However, I do ask you to get in touch with us after three months if you have not completed your revision, to update us on the status. Please also contact us as soon as possible if similar work is published elsewhere.

I also suggest that you carefully adhere to our guidelines for publication in your next version, including our new requirements for supplemental data (see also below) to speed up the pre-acceptance process in case of a positive outcome.

I look forward to seeing a revised form of your manuscript as soon as possible.

***** Reviewer's comments *****

Referee #1 (Comments on Novelty/Model System):

Please see my comments to be sent to the authors below.

Referee #1 (Remarks):

Bolino et al., "Niacin-mediated Tace activation ameliorates CMT neuropathies with focal hypermyelination"

Charcot-Marie-Tooth (CMT) diseases collectively represent a relatively common form of neuropathies for which there is no available treatment. The study of Bolino and colleagues concentrates on modulation of Neuregulin-1/ErbB2/B3-PI3K-Akt signaling pathway activity via TACE activity as a candidate treatment for some forms of neuropathy. While the presented data are potentially interesting, there are multiple questions that should be addressed. In particular, the authors should make an added effort in performing behavioral/ electrophysiological characterization of the possible improvements in neuropathy related phenotypes in characterized CMT models (CMT4B1 and HNPP). This is a critical point in order to link the observed structural improvement with potential positive therapeutical outcome.

I have the following specific comments:

The authors show that the Nrg1 type III expression and activity (as measured by pErbB2, pAkt and pErk; Figure 1) is not changing in Mtmr2 KO sciatic nerves or in DRG cultures from these animals. How does this support the proposed mechanism that modulation of TACE (which is a negative regulator of Nrg1 type III mediated myelination) may ameliorate hypermyelination in Mtmr2 KO model?

What happens to Mtmr2 KO SC/DRG neuron co-cultures treated with LV - NRG1-type III (Figure 2E) in terms of number of Mbp segments? This quantification should be presented also in Figures 3B (rhTACE treatment) and 3D (Niacin treatment). If the authors do observe a change in the number

of Mbp segments under these conditions, they should discuss this observation in the context of their hypothesis that the modulation of Nrg1 type III mediated signaling may ameliorate hypermyelination.

The data presented in Figure 4 suggest that in addition to decreasing the number of outfoldings present in Mtmr2 KO nerves, the Mtmr2 KO/ Nrg1 (III) +/- mice are hypomyelinated (as expected based on the work presented in Michailov et al., 2004). Hence the authors should reconsider their conclusion that the "downregulation of NRG1 type III in the Mtmr2-/- genetic background can benefit focal hypermyelination and myelin outfoldings without impairing myelination".

It is somehow surprising that the observed small change (increased g-ratio) in Pmp22+/- mice treated with Niaspan (Figure 7D) is statistically significant, while the change with the same amplitude - but going in the opposite direction - in Mtmr2 KO mice treated with Niaspan (Figure 6B) is not. The authors should provide an explanation for this discrepancy.

Niaspan treatment is targeting cholesterol and triglyceride levels in humans. The observed absence of changes in cholesterol and triglyceride levels in Mtmr2 mice (Figure EV2) treated with Niaspan should therefore be discussed.

Minor comments:

Page 4: "Binding of Nrg1 to their cognate receptors ErbB2/B3 on Schwann cell plasma membrane activates the PI3K-AKT1 signaling pathway" - it is not known which Akt isoform is predominantly expressed/activated in Schwann cells so it would be more appropriate to use "PI3K-AKT". The same is also true for page 11 - discussion.

Page 6: "Recent studies have hypothesized that myelin outfoldings might be PIP3-dependent and that they are formed as a consequence of the PI3K/AKT pathway activation (Goebbels et al, 2012)". Only one study is cited so the sentence should read: "A recent study....".

Page 6, referring to Figure 1D on page 22: how many "n" were analyzed and please define what one "n" means.

Figures 1D, 1E but also all other figures with pAkt Western blot. Why did the authors select to only evaluate the phosphorylation of S473 (mTORC2 dependent) and not Thr308 (PDK1 dependent)?

The number of independent experiments should be consistently specified in all figure legends - e.g. in legend Figure 2E (page 23), Figure 3B (page 24) only the number of coverslips is indicated.

In legend Fig 3D (page 25): correct the p value (probably $p=0.037$).

Figure 4A. How do the authors explain that Mtmr2 KO and Mtmr2 KO; Nrg1 (III) +/- have the same number of fibers by nerve? It was shown previously (Taveggia et al., 2005) that the Nrg1 (III) +/- has less myelinated fibers and more unmyelinated fibers. Therefore, one would expect in regard to this result that the Mtmr2 KO / Nrg1 (III) +/- will have less myelinated fibers. Could the authors define "fibers" (only myelinated or both myelinated and unmyelinated fibers)?

Figure 4C. Nrg1(III) in figure labeling should be replaced by Nrg1(III) +/- . Is the relative intensity a measurement of pAkt or pAkt normalized to the total Akt and Calnexin?

Figure 6 page 28: "Note that myelin outfoldings in adult WT nerves are only occasionally observed (0.08% as a mean of n=8 nerves)." The data are not shown in the figure but described in the legend.

Since some of the referred papers describe a role of mTOR in developing tomacula and other myelin abnormalities, the authors should discuss the advantage of treating their models with Niaspan rather than with rapamycin. Also, the authors should address in the discussion the observation that Niaspan is rescuing the tomacula defect in the Pmp22+/- mice but not in the Pmp22 KOs.

Referee #2 (Comments on Novelty/Model System):

They have used ex vivo systems and mouse models. Some behavioral analysis could apparently not be done, but the authors provide a reasonable explanation for this.

Referee #2 (Remarks):

The authors present an exciting study and show that a FDA-approved drug is able to restore normal myelination in mouse models of Charcot Marie Tooth disease. The authors present evidence that the drug's action is through activation of the protease TACE, through its cleavage of neuregulin and subsequent changes in ErbB signaling.

The study is novel, well written and uses a wide range of methods from in vitro to in vivo analysis. The study will be highly interesting not only to scientists studying myelination and CMT diseases but also to scientists studying cell surface proteolysis in different organs, in particular through the TACE protease. The results provide a solid basis for further preclinical testing of Niaspan in myelination disorders.

A major weakness of the current version of the manuscript is that it fails to finally prove that the mechanism of action is indeed through TACE. The authors appear to have all tools at hand to finally resolve this issue.

The following points need to be addressed to improve the manuscript.

Major points

1. Hypermyelination in TACE-deficient mice/explants may be mechanistically different from myelin outfold. Thus, myelin effect of niacin-treatment of TACE KO cultures may be different from Mtmr2-KO cultures. As a final proof that the niacin effect in Mtmr2 KO cultures is truly mediated by TACE, TACE needs to be knocked-out/down in Mtmr2 KO cultures, at least in the neurons, and then be treated with niacin. Ideally, this would even be done in mice. A similar (ex vivo) approach needs to be done for the Vim KO cultures.
2. The references in the manuscript are missing.
3. Figure 1C: At P20 and P60 the minor bands of inactive TACE are hardly visible, making it difficult to judge whether there is indeed an increase in the Mtmr2^{-/-} mice. The authors need to show statistics for these differences. What is the n number? Is there a statistically significant change?
4. Figure 1: If there are no changes in ErbB signaling between genotypes, why are there changes in myelination that may be corrected by TACE activation? Shouldn't ErbB signaling be increased when TACE is less mature/active? This needs to be clarified as it is the basis for the rationale to use a TACE activator.
5. Be careful with calling Niaspan a TACE activator. The name implies a direct binding to TACE, which does not seem to have been shown yet. The mechanism of action appears to be unclear. The drug is mostly used for reducing hyperlipidemic conditions. While it was found to enhance TACE expression in particular cells, it is unclear whether this happens directly or indirectly.
6. How can Niaspan help in all types of CMT disease, given that TACE blocks NRG signaling, whereas hypermyelination is only seen in some CMT forms? In the introduction the authors suggest a general approach to ALL CMT forms.
7. Figure 3D: how was TACE activity measured? Synthetic peptides alone are not a specific measure of TACE activity, unless when they are well controlled such as with TACE knock-down/out experiments.

Minor points:

8. In the introduction of discussion you need to mention and discuss the previous, partly diverging, findings from Fleck et al (J Neurosci 2013) regarding the involvement of ADAM17/TACE in

NRG1 processing.

9. Please indicate in the blots/legends what the loading control CLX is (calnexin?).

10. Figure 2E: label the outfolds in the picture to make it more easily understandable to scientists outside of the field. And is p-AKT reduced? Provide a quantification.

Referee #4 (Remarks):

This is a potentially important paper that reports on a novel experimental therapy in the treatment of CMT neuropathies. The key message of the paper is that niacin efficiently reduces myelin overgrowth profiles in several mice models of CMT. In addition, the authors provide genetic data that niacin targets TACE-dependent neuregulin processing. Furthermore, the authors confirm in genetic experiments that blocking neuregulin-dependent signalling indeed rescues the myelin overgrowth phenotype.

The study is interesting, technically well performed, and of high clinical relevance as niacin is a FDA approved drugs.

Major points:

1) The study is entirely based on morphological readouts - myelin outfoldings. Does the rescue of myelin outfoldings improve the performance of peripheral nerves? If electrophysiological data cannot be presented, at least an analysis of the axonal degeneration should be presented.

2) Does niacin application lead to differences in the number of myelinated axons? Looking at the EM images it does not look like there are differences, but this should be quantified. Is there any evidence from clinical trials that nicacin causes hypomyelinating neuropathies after long-term treatment?

Text:

Page 5/6: Both paragraph start with the same sentence.

Page 6: inactive form of Tace, please explain

Page 7: rhTACE, please explain

Page 8: preclinical study, a bit too much, the endpoint is histology

Discussion: The discussion on myelin outfoldings and tomacula should be extended. How are they generated and how do they induce axonal damage? It should also be mentioned that they are a normal feature of developing myelin in the CNS.

Figures:

The figures are in general too complex. The authors should consider putting some of the data into the supplements. For example Figure 2: Main finding is shown in Figure 2E. The data shown in A-D are controls of the efficiency of the shRNA.

Often the number of experiments is not shown. This should be corrected for all figures.

Figure 1C: the quantification is difficult to understand. The band intensities look very similar. If quantitative data is shown, this should be quantified in more experiments showing standard deviations.

Figure 4C: There is lot of variation in the data. It looks like Akt activation could be similar.

Enclosed please find the revised version of our manuscript entitled: **Niacin-mediated Tace activation ameliorates CMT neuropathies with focal hypermyelination.**

First of all, we would like to thank the Reviewers and the Editors for their comments. According to Reviewer's and Editor's suggestions, we changed our manuscript as follows (please note text changes in yellow):

Referee #1:

Charcot-Marie-Tooth (CMT) diseases collectively represent a relatively common form of neuropathies for which there is no available treatment. The study of Bolino and colleagues concentrates on modulation of Neuregulin-1/ErbB2/B3PI3K-Akt signaling pathway activity via TACE activity as a candidate treatment for some forms of neuropathy. While the presented data are potentially interesting, there are multiple questions that should be addressed. In particular, the authors should make an added effort in performing behavioral/ electrophysiological characterization of the possible improvements in neuropathy related phenotypes in characterized CMT models (CMT4B1 and HNPP). This is a critical point in order to link the observed structural improvement with potential positive therapeutical outcome.

Answer: Unfortunately the phenotype of the *Mtmr2*^{-/-} mouse, a model of CMT4B1, is relatively mild. This mutant is clinically normal. At the neurophysiological examination, nerve conduction velocity decreases of 6-8 m/s in the KO as compared to controls only at 6 months of age. Moreover, at 6 months of age, by performing footprint gait analysis, we found that KO mice display a modest increase in the base length and strides as compared to controls (an increase of 10-20%). All other tests performed including rotarod analysis and grid analysis were normal (*Bolino et al., JCB 2004*). Before 6 months of age, these outcome measures are not applicable.

The *Pmp22*^{+/-} phenotype is also relatively mild and characterized by the presence of tomacula in the nerve (*Adlkofer, J Neurosci 1997*). europhysiology is almost normal in this mutant. Axonal degeneration is not a feature of this model until 12-14 months of age. Demyelination and degeneration of tomacula are evident starting from 6 months. The limitation of these two models has been now discussed in more detail in the new version of the Discussion (pages 12-16).

Finally, conduction blocks have been described in *Pmp22*^{+/-}-nerves at 2 months of age by Dr. Jun Li using a complex neurophysiological set up (*Bai et al., J Neurosci 2010*). We tried to reproduce the experimental setting reported in this paper but we did not succeed in detecting conduction blocks in *Pmp22*⁺/adult sciatic nerves.

Given the lack of functional outcome measures for these models, we tried as much as possible to soften our conclusions in the revised version of this paper.

I have the following specific comments:

1-The authors show that the Nrg1 type III expression and activity (as measured by pErbB2, pAkt and pErk; Figure 1) is not changing in Mtmr2 KO sciatic nerves or in DRG cultures from these animals. How does this support the proposed mechanism that modulation of TACE (which is a negative regulator of Nrg1 type III mediated myelination) may ameliorate hypermyelination in Mtmr2 KO model?

Answer: In this paper, we propose that the modulation of Tace activity and the downregulation of the Nrg1 type III pathway could be beneficial to CMT neuropathies characterized by focal hypermyelination independently of the pathomechanism underlying each CMT clinical sub-type, as Nrg1 type III is one the main signaling regulating myelination in the PNS. Nevertheless, we analyzed basal level of activation of this pathway in all the CMT model used in this study to correlate this pathway, if altered, with the observed amelioration.

To answer to reviewer's concern, we further investigated ErbB2 receptor phosphorylation levels in

Mtmr2^{-/-}-at P2 using additional animals and pools of nerves. Using at least n=8 pools per genotype of sciatic nerves (each pool is including nerves from 6/7 animals), we found that ErbB2 receptor phosphorylation was significantly increased in *Mtmr2*^{-/-}-nerves at P2. Moreover, using myelin-forming Schwann cell/DRG neuron co-cultures after 4 days of ascorbic acid treatment from *Mtmr2*^{-/-}-embryos and control, we found that the increase ErbB2 receptor phosphorylation was also recapitulated *in vitro* in mutant embryos. We show this result in the new version of Figure 1, and in the text, pages 6-7, Results.

Increased ErbB2 receptor phosphorylation is also a feature of CMT4C caused by mutations in the *SH3TC2* gene. The SH3TC2 protein is thought to regulate ErbB2 receptor trafficking and recycling at the Schwann cell plasma membrane (Stendal *et al.*, *Brain* 2010). Interestingly, in addition to redundant basal lamina, myelin outfoldings have also been observed in some CMT4C patient sural nerve biopsies (Dr. Previtali and Dr. Fabrizi, personal communication). Concerning *Mtmr2*^{-/-}-and CMT4B1, we can speculate that loss of Mtmr2 might impair ErbB2 receptor trafficking at the level of the endocytic pathway, as a consequence of altered levels of PtdIns(3,5)P₂. ErbB2 receptors may still signal in endosomes and locally activate downstream effectors relevant for myelination (see page 7).

2-What happens to Mtmr2 KO SC/DRG neuron co-cultures treated with LV NRG1-type III (Figure 2E) in terms of number of Mbp segments? This quantification should be presented also in Figures 3B (rhTACE treatment) and 3D (Niacin treatment). If the authors do observe a change in the number of Mbp segments under these conditions, they should discuss this observation in the context of their hypothesis that the modulation of Nrg1 type III mediated signaling may ameliorate hypermyelination.

Answer: In Figure 2E, we already reported a mild decrease of myelination in *Mtmr2*^{-/-}-cultures transduced with LV carrying shRNA Nrg1, similarly to wild type cultures, see Legend of Fig 2E, total number of Mbp fibers: in NT, 382; in 12.5% LV, 368 and in cultures transduced with 25% LV, 327. The downregulation of Nrg1 expression is leading to a mild reduction of the myelin content as compared to the extent of the rescue of myelin outfoldings. We used the downregulation of Nrg1 *in vitro* and *in vivo* to provide proof of principle data. On the other hand, the modulation of Tace activity using the pharmacological approach is intended to ameliorate hypermyelinating neuropathies without affecting myelin thickness, either *in vitro* or *in vivo*. Consistent with this, we measured Mbp positive fibers in treated and not treated *Mtmr2*^{-/-}-cultures (either with Niacin or rhTACE) and we did not observe significant differences (see Legends Fig. 3B and 3D of the paper, pages 39-40).

3-The data presented in Figure 4 suggest that in addition to decreasing the number of outfoldings present in Mtmr2 KO nerves, the Mtmr2 KO/Nrg1 (III) +/-mice are hypomyelinated (as expected based on the work presented in Michailov et al., 2004). Hence the authors should reconsider their conclusion that the "downregulation of NRG1 type III in the Mtmr2/-genetic background can benefit focal hypermyelination and myelin outfoldings without impairing myelination".

Answer: We agree with the Reviewer and we apologize for this conclusion. We removed this sentence at the end of the paragraph and modified the conclusion, as now in Page 8.

4-It is somehow surprising that the observed small change (increased g-ratio) in Pmp22+/-mice treated with Niaspan (Figure 7D) is statistically significant, while the change with the same amplitude -but going in the opposite direction in Mtmr2 KO mice treated with Niaspan (Figure 6B) is not. The authors should provide an explanation for this discrepancy.

Answer: In Fig. 6B, mean g-ratio values are similar between the three groups: WT 0.67, *Mtmr2*^{-/-}-Niaspan 0.67, and *Mtmr2*^{-/-}-saline 0.68.

In Fig. 7D, instead, WT mean g-ratio value is 0.69, *Pmp22*^{+/-}-Niaspan-treated 0.68, and *Pmp22*^{+/-}-saline-treated is 0.665 (now reported in the Legend of Fig. 7, page 44). In the previous version of the paper, we approximated this latter value (*Pmp22*^{+/-}-saline-treated) at 0.67. On the basis of the sampling (please see the total number of fibers), SD/SEM and the same statistical analysis between the two experiments reported in Fig. 7D and 6B, Anova repeated measures, the differences among

mean g-ratio values are statistically significant in Fig. 7D whereas in Fig 6B are not.

5-Niaspan treatment is targeting cholesterol and triglyceride levels in humans. The observed absence of changes in cholesterol and triglyceride levels in Mtmr2 mice (Figure EV2) treated with Niaspan should therefore be discussed.

Answer: See now discussion page 16.

In human, when administered at a dose of 15-20mg/day Niacin/nicotinic acid acts as a vitamin. At supraphysiological doses it exerts a variety of pharmacological effects. The most known FDA-approved effect is the antilipidemic effect. Niacin and the extended release formulation of Niacin, Niaspan, have been shown to lower plasma levels of FFA, LDL-C, VLDL, cholesterol, and TG, whereas HDL increases (*Bodor and Offermanns British J of Pharmacology, 2008*). Gram ranges have anti-lipidemic effects in human. In the mouse, 0.3% to 3% (w/w) dose ranges have antilipidemic effects, which correspond to 100-330 mg of compound per mouse (body weight of 30 g).

The dosage we administered to activate Tace and evaluate the effect on myelination was of 160 mg/Kg, which corresponds to 5 mg in a 30 g mouse. At this dosage, Niacin has no antilipidemic effect. Moreover, Niacin was shown to have antilipidemic efficacy not in physiological but in dyslipidemic conditions, as for the transgenic line *Tg(CETP/Ldlrtm1)*, having high plasma levels of LDL (*Lauring et al., Science Translational Medicine, 2012*).

Bioclinical analyses in Niaspan treated models have been performed to rule out metabolic toxicity due to long-term daily administration of Niaspan in these mutants.

Minor comments: Page 4: "Binding of Nrg1 to their cognate receptors ErbB2/B3 on Schwann cell plasma membrane activates the PI3K-AKT1 signaling pathway" -it is not known which Akt isoform is predominantly expressed/activated in Schwann cells so it would be more appropriate to use "PI3K-AKT". The same is also true for page 11 -discussion.

Answer: we removed AKT1 throughout the text.

Page 6: "Recent studies have hypothesized that myelin outfoldings might be PIP3-dependent and that they are formed as a consequence of the PI3K/AKT pathway activation (Goebbels et al, 2012)". Only one study is cited so the sentence should read: "A recent study....".

Answer: the paper by Domènech-Estévez E et al, J Neurosci, April 2016 reports that *in vivo* the overactivation of Akt in the nerve results, among others, in the formation of occasional myelin outfoldings/infoldings and tomacula. We added this reference in this sentence and we modified this sentence (results page 6).

Page 6, referring to Figure 1D on page 22: how many "n" were analyzed and please define what one "n" means.

Answer: this has been specified in Legend of Fig EV3 of the revised version, page 48.

Figures 1D, 1E but also all other figures with pAkt Western blot. Why did the authors select to only evaluate the phosphorylation of S473 (mTORC2 dependent) and not Thr308 (PDK1 dependent)?

Answer: We and others often observed the activation of a negative feedback loop in Schwann cells between mTOR and molecules upstream of PI3K, thus producing opposite effects on AKT phosphorylation between AKT S473 and AKT T308 (*reviewed in Norrmen et al., Biochem Soc Transactions 2013 and in Laplante et al., Cell 2012*). Thus, to monitor AKT activation, S473 not T308 has been investigated in several models *in vivo* in the nerve.

The following studies report AKTS473 or T308 changes in the nerve:

a) Goebbels et al., J Neurosci 2010 and EMBO Mol Med 2012: mutant nerves lacking Pten in Schwann cells with higher PIP3 levels have increased S473-AKT phosphorylation levels (and increased myelin thickness). AKT phosphorylation at T308 was not investigated in this study.

- b) Cotter et al, Science 2010, demonstrated a correlation between Dlg1 and phosphorylation of AKT at S473 both *in vivo* and *in vitro*.
- c) Nosedà et al., J Neurosci 2013: mutant nerves lacking Dlg1 in Schwann cells display increased levels of S473-AKT phosphorylation (and increased myelin thickness). Mutant nerves lacking Ddit4 have increased pS6 and S473 AKT phosphorylation (and increased myelin thickness), but lower AKT T308 phosphorylation levels.
- d) in Sherman et al., J Neurosci 2012: mutant nerves lacking mTOR in Schwann cells display almost normal levels of S473 (and decreased myelin thickness), whereas T308 phosphorylation is increased.
- e) in Norrmen et al., Cell Reports 2014: mutant nerves lacking both Raptor and Rictor display decreased levels of S473 (and decreased myelin thickness) and increased levels of T308 phosphorylation.
- f) in Fledrich R et al., Nat Medicine 2014, the authors propose that the unbalance between Akt and Erk activation is at the basis of CMT1A pathogenesis and that administration of soluble Nrg1 to CMT1A models ameliorates the phenotype by restoring levels of Akt activation. Only p-Akt at S473 was investigated.

The number of independent experiments should be consistently specified in all figure legends -e.g. in legend Figure 2E (page 23), Figure 3B (page 24) only the number of coverslips is indicated.

Answer: In Fig. 2E we performed one experiment for sh#1 and one experiment for sh#2, which produced similar results. Only the experiment relative to shRNA #1 is shown.

In Fig. 3B, one experiment has been performed using at least n=10 different coverslips/DRGs.

In legend Fig 3D (page 25): correct the p value (probably p=0.037).

Answer: the typing error has been corrected.

Figure 4A. How do the authors explain that Mtmr2 KO and Mtmr2 KO; Nrg1 (III) +/- have the same number of fibers by nerve? It was shown previously (Taveggia et al., 2005) that the Nrg1 (III) +/- has less myelinated fibers and more unmyelinated fibers. Therefore, one would expect in regard to this result that the Mtmr2 KO / Nrg1 (III) +/- will have less myelinated fibers. Could the authors define "fibers" (only myelinated or both myelinated and unmyelinated fibers)?

Answer: in Taveggia et al., 2005 the number of fibers has been assessed by ultrastructural analysis. To evaluate the number of myelin outfoldings in *Mtmr2*^{-/-} sciatic nerve sections, we performed semithin section analysis. We counted the total number of myelinated fibers (and myelin outfoldings) in the entire nerve section. We changed in "myelinated fibers" the histogram of Fig. 4A. We can only speculate that the downregulation of Nrg1 type III in the *Mtmr2*^{-/-} mutant background may not entirely recapitulate the *Nrg1*(III)^{+/-} phenotype: *Mtmr2*^{-/-};*Nrg1*(III)^{+/-} nerves have a similar number of myelinated fibers as compared to *Mtmr2*^{-/-}. Finally, *Mtmr2*^{-/-};*Nrg1*(III)^{+/-} mutant nerves have intermediate g-ratio values between *Mtmr2*^{-/-} and *Nrg1*(III)^{+/-} single mutants.

Figure 4C. Nrg1(III) in figure labeling should be replaced by Nrg1(III) +/- . Is the relative intensity a measurement of pAkt or pAkt normalized to the total Akt and Calnexin?

Answer: the labeling has been corrected. Relative intensity is: total Akt normalized to the loading-calnexin, and then phosphorylation of Akt is normalized over this ratio. We specified the quantification made in the legend.

Figure 6 page 28: "Note that myelin outfoldings in adult WT nerves are only occasionally observed (0.08% as a mean of n=8 nerves)." The data are not shown in the figure but described in the legend.

Answer: we removed this sentence from the legend. We ment that in wild type nerves myelin outfoldings are almost absent.

Since some of the referred papers describe a role of mTOR in developing tomacula and other myelin abnormalities, the authors should discuss the advantage of treating their models with Niaspan

rather than with rapamycin. Also, the authors should address in the discussion the observation that Niaspan is rescuing the tomacula defect in the *Pmp22*^{+/-}-mice but not in the *Pmp22* KO's.

Answer: these two issues are now in the Discussion pages 15-17.

Referee #2 (Comments on Novelty/Model System): They have used ex vivo systems and mouse models. Some behavioral analysis could apparently not be done, but the authors provide a reasonable explanation for this.

Referee #2 (Remarks): The authors present an exciting study and show that a FDA-approved drug is able to restore normal myelination in mouse models of Charcot Marie Tooth disease. The authors present evidence that the drug's action is through activation of the protease TACE, through its cleavage of neuregulin and subsequent changes in ErbB signaling. The study is novel, well written and uses a wide range of methods from in vitro to in vivo analysis. The study will be highly interesting not only to scientists studying myelination and CMT diseases but also to scientists studying cell surface proteolysis in different organs, in particular through the TACE protease. The results provide a solid basis for further preclinical testing of Niaspan in myelination disorders. A major weakness of the current version of the manuscript is that it fails to finally prove that the mechanism of action is indeed through TACE. The authors appear to have all tools at hand to finally resolve this issue. The following points need to be addressed to improve the manuscript.

Major points

1. Hypermyelination in TACE-deficient mice/explants may be mechanistically different from myelin outfoldings. Thus, myelins effect of niacin-treatment of TACE KO cultures may be different from *Mtmr2*-KO cultures.

As a final proof that the niacin effect in *Mtmr2* KO cultures is truly mediated by TACE, TACE needs to be knocked-out/down in *Mtmr2* KO cultures, at least in the neurons, and then be treated with niacin. Ideally, this would even be done in mice. A similar (ex vivo) approach needs to be done for the *Vim* KO cultures.

Answer: We produced lentiviral vectors carrying three different shRNAs targeting *Tace* as well as one shRNA against a Scramble sequence to be used as control. These lentiviral vectors have been validated and extensively used by Dr. Taveggia as reported in *La Marca et al, Nature Neuroscience 2011*. We validated the lentiviral production by transducing isolated rat Schwann cells, which express high levels of *Tace* (qRT PCR analysis, Figure below panel A and novel Fig EV4). We also transduced wild type Schwann cell/DRG neuron co-culture explants using *Tace* shRNA #1 LV and Scramble LV. Explants transduced with shRNA #1 *Tace* produced more myelin segments as compared to Scramble LV-transduced explants, thus suggesting that also in these settings the downregulation of neuronal *Tace* expression results in an increase of myelination (*La Marca et al., Nat Neurosci 2011*) (Figure below panel B and the novel EV4 Figure).

Then, we transduced Schwann cell/DRG neuron co-culture explants from *Mtmr2*^{-/-}-mice using *Tace* shRNA #1 or Scramble shRNA and treated or not-treated using 5mM Niacin. As expected, Niacin significantly rescued myelin outfoldings in *Mtmr2*^{-/-}-treated explants as compared to *Mtmr2*^{-/-}-not-treated (both transduced with Scramble shRNA). On the other hand, Niacin did not rescue myelin outfoldings in *Mtmr2*^{-/-}-cells transduced using *Tace* shRNA 1 as compared to either *Mtmr2*^{-/-}-Scramble or *Mtmr2*^{-/-}-*Tace* shRNA #1 both not-treated explants. These data suggest that Niacin acts through *Tace* to improve myelination in *Mtmr2*^{-/-}. These results are shown in the novel EV4 Figure and text page 8.

We also established from *Vim*^{-/-}-and *Vim*^{+/-}-embryos. As expected and as already reported (*Triolo et al, Development 2012*) *Vim*^{-/-}-explants were more myelinated than *Vim*^{+/-}-ones and Niacin was able to restore myelination to normal levels in *Vim*^{-/-}-cultures (Panel C figure below and Figure 5 of this manuscript). However, *Vim*^{-/-}-cultures transduced using shRNA #1 *Tace* produced significantly less myelin segments as compared to either *Vim*^{-/-}-scramble LV or *Vim*^{+/-}-scramble LV (Figure below panel C: as an example we show the reconstruction of one DRG explant per condition stained using anti-Mbp antibodies to detect myelin segments; *Vim*^{-/-}-scramble n=6 coverslips/DRGs; *Vim*^{+/-}-

scramble n=10; *Vim*^{-/-}-scramble and Niacin-treated n=8; *Vim*^{-/-}-shRNA #1 Tace n=16; representative of two independent experiments). Thus, as the downregulation of Tace in *Vim*^{-/-}-has *per se* a negative effect on myelination at least in the acute setting of these *ex vivo* experiments, we did not further analyze *Vim*^{-/-}-shRNA#1 Tace infected DRGs and treated with Niacin.

In *Vim*^{-/-}-explants Nrg1 type III levels are increased and we already reported that Vimentin and Tace acts synergistically to control myelination (double heterozygous mice *Vim*^{+/-}; *Tace*^{+/-}-are hypermyelinated, *Triolo et al., Development 2012*). These new results suggest that loss of both Vimentin and Tace is detrimental for myelination. To further explore the specificity of Niacin on Tace in a Vimentin-null background, an *in vivo* approach should be undertaken as suggested by the reviewer by generating double null mice lacking both Vimentin and Tace (conditionally in neurons, as total KOs are lethal), which would take months and is beyond the intention of this manuscript.

2. The references in the manuscript are missing.

Answer: we do apologize for this mistake.

3. Figure 1C: At P20 and P60 the minor bands of inactive TACE are hardly visible, making it difficult to judge whether there is indeed an increase in the *Mtmr2*^{-/-} mice. The authors need to show statistics for these differences. What is the n number? Is there a statistically significant change?

Answer: Western blot images of Figure 1 panels A-C have been enlarged to appreciate the bands referring to active and inactive Tace isoforms. To better quantify Tace expression changes in *Mtmr2*^{-/-} nerves, we performed several western blot analyses using nerves at P20 and at P60 or 5 months (adult nerves). To this aim, we obtained from Abcam a new lot number of the ADAM17 antibody, which is different from the one we used to generate data shown in the previous version of the paper (in 2014, Abcam lot# 812280, whereas now in 2016, Abcam lot #GR263624-1). The new lot number was not efficient as the previous in detecting Tace, particularly the inactive Tace isoform (110 KDa), which is also limiting as it is the less abundant one. Unfortunately, we were not able to detect a significant increase of Tace expression in *Mtmr2*^{-/-} nerves at P20. Please see the western blots below as examples. Please also note that, to confirm the specificity of the new Tace antibody, lysates from *Tace*^{Fl/Fl}; *P0*Cre mice, in which only axonal Tace is expressed, were always loaded as controls together with WT at P10-the time point at which expression of the inactive Tace isoform is not yet downregulated.

We also performed two independent experiments to quantify Tace expression levels in adult *Mtmr2*^{-/-} nerves. We observed a modest but non significant increase of the active Tace isoform (80 KDa) expression in *Mtmr2* mutant nerves, whereas the upper band at 110 KDa, which correspond to the inactive isoform, was not well resolved. This result is now shown in Figure 1, panel D.

4. *Figure 1: If there are no changes in ErbB signaling between genotypes, why are there changes in myelination that may be corrected by TACE activation? Shouldn't ErbB signaling be increased when TACE is less mature/active? This needs to be clarified as it is the basis for the rationale to use a TACE activator.*

Answer: please refer to the answer to Rev#1, specific comments, point 1.

5. *Be careful with calling Niaspan a TACE activator. The name implies a direct binding to TACE, which does not seem to have been shown yet. The mechanism of action appears to be unclear. The drug is mostly used for reducing hyperlipidemic conditions. While it was found to enhance TACE expression in particular cells, it is unclear whether this happens directly or indirectly.*

Answer: we do agree and we modified the text of the paper according to Reviewer's comment.

6. *How can Niaspan help in all types of CMT disease, given that TACE blocks NRG signaling, whereas hypermyelination is only seen in some CMT forms? In the introduction the authors suggest a general approach to ALL CMT forms.*

Answer: we do agree and we modified the sentence in the introduction, page 4 and in the abstract, page 2.

7. *Figure 3D: how was TACE activity measured? Synthetic peptides alone are not a specific measure of TACE activity, unless when they are well controlled such as with TACE knock-down/out experiments.*

Answer: we detailed the protocol used in the material and methods section, pages 20 and 21.

Minor points:

8. *In the introduction of discussion you need to mention and discuss the previous, partly diverging, findings from Fleck et al (J Neurosci 2013) regarding the involvement of ADAM17/TACE in NRG1 processing.*

Answer: we modified the text accordingly, now in the Discussion page 13.

9. *Please indicate in the blots/legends what the loading control CLX is (calnexin?).*

Answer: we specified the abbreviation of the loading controls used in each western blot.

10. *Figure 2E: label the outfoldings in the picture to make it more easily understandable to scientists outside of the field. And is p-AKT reduced? Provide a quantification.*

Answer: we labeled outfoldings in the Mbp panel of confocal images. P-Akt was quantified in the legend as a ratio.

Referee #4 (Remarks):

This is a potentially important paper that reports on a novel experimental therapy in the treatment of CMT neuropathies. The key message of the paper is that niacin efficiently reduces myelin overgrowth profiles in several mice models of CMT. In addition, the authors provide genetic data that niacin targets TACE-dependent neuregulin processing. Furthermore, the authors confirm in

genetic experiments that blocking neuregulin-dependent signalling indeed rescues the myelin overgrowth phenotype.

The study is interesting, technically well performed, and of high clinical relevance as niacin is a FDA approved drugs.

Major points: 1) The study is entirely based on morphological readouts -myelin outfoldings. Does the rescue of myelin outfoldings improve the performance of peripheral nerves? If electrophysiological data cannot be presented, at least an analysis of the axonal degeneration should be presented.

Answer: unfortunately, axonal degeneration is not a pathological feature of *Mtmr2*^{-/-} and *Pmp22*^{+/-} models. In particular, in *Mtmr2*^{-/-} sciatic nerves occasional degenerating axons have been observed only at 6 months of age (Bolino *et al.*, *Journal of Cell Biology* 2004). In *Pmp22*^{+/-} mutants, tomacula leading to axonal displacement can be observed at 10 months of age. Demyelination defined as the degeneration of hypermyelinated structures tomacula (not axons) has been reported in sciatic nerves of 15 months-old *Pmp22*^{+/-} mice (Aldkofer *et al.*, *J Neurosci* 1997). This issue has been now reported in the Discussion, pages 12-15.

2) Does niacin application lead to differences in the number of myelinated axons? Looking at the EM images it does not look like there are differences, but this should be quantified. Is there any evidence from clinical trials that nicacin causes hypomyelinating neuropathies after long-term treatment?

Answer: In Fig. 6B we already reported that the number of myelinated fibers is not changed between *Mtmr2*^{-/-}-saline and *Mtmr2*^{-/-}-Niaspan treated mice. We included this information also in Fig. 5 (*Vim* model) and in Fig. 7 (*Pmp22* model).

Finally, Niacin is not reported in the literature to be causative of neuropathies. The only evidence is related to statins, another antilipidemic drug acting through a different molecular mechanism.

Text: Page 5/6: Both paragraph start with the same sentence.

Answer: we removed the first sentence at the beginning of the paragraph in page 6, now page 7. We started with similar sentences to recall the proposed strategy when reporting the *ex vivo* and *in vivo* results.

Page 6: inactive form of Tace, please explain

Answer: it means “Tace not yet processed by furin cleavage”. For reference: Gooz M. *ADAM-17: the enzyme that does it all*. *Crit Rev Biochem Mol Biol*. 2010 Apr;45(2):146-69. We modified the text accordingly, page 5.

Page 7: rhTACE, please explain

Answer: we specified recombinant human TACE, page 10.

Page 8: preclinical study, a bit too much, the endpoint is histology Answer: we removed “preclinical” from this sentence, and we used instead “*in vivo*”, page 8.

Discussion: The discussion on myelin outfoldings and tomacula should be extended. How are they generated and how do they induce axonal damage? It should also be mentioned that they are a normal feature of developing myelin in the CNS.

Answer: we revised the discussion accordingly, pages 12-17.

Figures: The figures are in general too complex. The authors should consider putting some of the data into the supplements. For example Figure 2: Main finding is shown in Figure 2E. The data

shown in A-D are controls of the efficiency of the shRNA.

Answer: we modified Figure 1 and we moved part of the western blots showing basal levels to EV1-3 figures.

Often the number of experiments is not shown. This should be corrected for all figures.

Answer: we specified the number of experiments throughout the text.

Figure 1C: the quantification is difficult to understand. The band intensities look very similar. If quantitative data is shown, this should be quantified in more experiments showing standard deviations.

Answer: please see Reviewer's 2 answer, point 3.

Figure 4C: There is lot of variation in the data. It looks like Akt activation could be similar.

Answer: indeed, the difference between *Mtmr2*^{-/-} and *Mtmr2*^{-/-}; *Nrg1*(III)^{+/-} is not significant. The levels of Akt phosphorylation in *Mtmr2*^{-/-}; *Nrg1*(III)^{+/-} nerves are similar to those of *Nrg1*(III)^{+/-} nerves.

We hope that you will find this manuscript sufficient for publication. Thank you in advance for your kind attention.

2nd Editorial Decision

15 September 2016

Thank you for the submission of your revised manuscript to EMBO Molecular Medicine. We have now received the enclosed reports from the referees that were asked to re-assess it. As you will see the reviewers are now globally supportive and I am pleased to inform you that we will be able to accept your manuscript pending the following final amendments:

- 1) Please take action on the remaining issues mentioned by reviewer 1.
- 2) While performing our pre-publishing quality control and image screening routines, we noticed a possible instance of image duplication in your figures, whereby two different panels appear too similar to be different. This issue prevents us from moving forward with your manuscript and we must therefore ask you to please provide us with the full source data set for these images, a corrected figure and an explanation of the occurrence.
- 3) Related to the above, we encourage the publication of source data, particularly for electrophoretic gels and blots, with the aim of making primary data more accessible and transparent to the reader. Would you be willing to provide a PDF file per figure that contains the original, uncropped and unprocessed scans of all or at least the key gels used in the manuscript (in addition to the case mentioned in item 2 above)? The PDF files should be labeled with the appropriate figure/panel number, and should have molecular weight markers; further annotation may be useful but is not essential. The PDF files will be published online with the article as supplementary "Source Data" files. If you have any questions regarding this just contact me.

Please submit your revised manuscript and the additional material requested within two weeks. I look forward to seeing a revised form of your manuscript as soon as possible.

***** Reviewer's comments *****

Referee #1 (Remarks):

Bolino et al., "Niacin-mediated Tace activation ameliorates CMT neuropathies with focal hypermyelination"

The revised version of the manuscript partially answered my comments and questions. In particular, the authors made an effort to establish that Nrg1/ErbB signaling is increased in *Mttr2* KO animals and that Niaspan/Niacin is acting through TACE to improve myelination. It is somehow frustrating that the authors were not able to provide any data demonstrating the functional recovery in the studied models. However, this limitation is now clarified in the text.

Minor comments:

Figure 1E and F, EV1-3:

The majority of analyzed proteins showed no changes (Nrg1, p-Akt, p-Erk), the only one with small increase was p-ErbB2. This observation could be substantiated by the data on p-ErbB3.

Figure 4C:

Add "Akt activation or phosphorylation" in the y axis legend of the histogram graph (Akt activation / relative intensity).

Figure EV4E:

Add the word "Niacin" in to the x axis legend (5mM Niacin).

Referee #2 (Remarks):

The authors have adequately addressed my previous concerns.

Referee #4 (Remarks):

The authors have done a good job in answering all of the questions raised by the reviewers. This is well performed analysis that will be of high interest.

2nd Revision - authors' response

22 September 2016

Enclosed please find the revised version of our manuscript entitled: **Niacinmediated Tace activation ameliorates CMT neuropathies with focal hypermyelination.**

We modified the paper as follows:

Reviewer #1.

Figure 1E and F, EV1-3:

The majority of analyzed proteins showed no changes (Nrg1, p-Akt, p-Erk), the only one with small increase was p-ErbB2. This observation could be substantiated by the data on p-ErbB3.

Answer: we were able to quantify the increase of ErbB2 receptor phosphorylation at P2 by performing several western blot experiments using at least 6-8 pools of nerves per genotype at P2. To have a sufficient amount of lysate to be loaded, each pool contains 6-7 animals (12-14 nerves) at this age. We could not repeat this experiment to detect phospho-ErbB3 using new lysates/animals. Please also note that the *Mttr2*^{-/-} born ratio is less than 25%.

In order to hybridize membranes already prepared and used for the ErbB2 detection (the anti-phospho ErbB2 used was a rabbit antibody), we looked for a monoclonal antibody recognizing mouse phosphorylated ErbB3 and produced in mouse, rat or goat. Unfortunately, there are no commercial monoclonal antibodies raised in these species which can recognize mouse ErbB3. They are usually made to specifically recognize the human antigen.

Figure 4C:

Add "Akt activation or phosphorylation" in the y axis legend of the histogram graph (Akt activation / relative intensity).

Answer: we modified the figure as suggested.

Figure EV4E:

Add the word "Niacin" in to the x axis legend (5mM Niacin).

Answer: we modified the figure as suggested.

We also checked and revised Figure EV3 panel A as suggested. We did a mistake whilst preparing the Figure panel and we included the same phospho-Akt blot to display both phospho-Akt and total Akt (only for wild-type samples).

We do apologise for this mistake. As it can be appreciated by the Supplementary material-raw data for this blot, total Akt is different from phospho-Akt for all the samples shown in the Figure. The lanes in between KO and WT in the film are referring to another experiment. Accordingly, since we cropped the film to prepare the Figure, we displayed the panels always separated from each other.

Regarding Figures, please note that we prepared our Figures in RGB mode. We tried to convert the profile in CMYK but we lost contrast and brightness. The submitted files are thus Tiff 300 dpi and RGB mode.

Finally, we uploaded as requested all the raw-source data for the blots displayed in the Figures.

We hope you will find our manuscript sufficient for publication.

Corresponding Author Name: Alessandra Bolino

Journal Submitted to: EMBO MOLECULAR MEDICINE

Manuscript Number: EMM-2016-06349